



# Changes in Surface Broadband Shortwave Radiation Budget during the 2017 Eclipse

Guoyong Wen[1,2], Alexander Marshak[1], Si-Chee Tsay[1], Jay Herman[1,3], Ukkyo Jeong[1,4], Nader Abuhassan[1,3], Robert Swap[1], Dong Wu[1]

[1]NASA/Goddard Space Flight Center, Code 613, Greenbelt, Maryland, 20771, USA
[2]GESTAR/Morgan State University, Baltimore, Maryland, 21251, USA
[3]JCET, University of Maryland Baltimore County, Baltimore, Maryland, 21250, USA
[4]ESSIC, University of Maryland, College Park, Maryland, 20742, USA

Correspondence to: Guoyong Wen (Guoyong.Wen@nasa.gov)

**Abstract.** While solar eclipses are known to greatly diminish the visible radiation reaching the surface of the Earth, less is known about the magnitude of the impact. We explore both the observed and modelled level of change in surface radiation during the eclipse of 2017. We deployed a pyranometer and Pandora spectrometer instrument to Casper, Wyoming and Columbia, Missouri to measure surface broadband shortwave (SW) flux and atmospheric properties during the 21 August 2017 solar eclipse event. We performed detailed radiative transfer simulations to understand the role of clouds in spectral and broadband solar radiation transfer in the Earth's atmosphere for the normal (non-eclipse) spectrum and red-shift solar spectra for eclipse conditions. The theoretical calculations showed that the non-eclipse-to-eclipse surface flux ratio depends strongly on the obscuration of solar disk and slightly on cloud optical depth. These findings allowed us to estimate what the surface broadband SW flux would be for non-eclipse conditions from observations during the eclipse and further to quantify the impact of the eclipse on the surface broadband SW radiation budget. We found that the eclipse caused local reductions of time-averaged surface flux of about 379 W m$^{-2}$ (50%) and 329 W m$^{-2}$ (46%) during the ~3 hours course of the eclipse at the Casper and Columbia sites, respectively. We estimated that the Moon's shadow caused a reduction of approximately 7-8% in global average surface broadband SW radiation. The eclipse has a smaller impact on absolute value of surface flux reduction for cloudy conditions than a clear atmosphere; the impact decreases with the increase of cloud optical depth. However, the relative time-averaged reduction of local surface SW flux during a solar eclipse is approximately 45% and it is not sensitive to cloud optical depth. The reduction of global average SW flux relative to climatology is proportional to the non-eclipse and eclipse flux difference in the penumbra area and depends on cloud optical depth in the Moon's shadow and geolocation due to the change of solar zenith angle. We also discuss the influence of cloud inhomogeneity on the observed SW flux. Our results not only quantify the reduction of the surface solar radiation budget but also advance the understanding of broadband SW radiative transfer under solar eclipse conditions.



## 1 Introduction

On 21 August 2017, a total solar eclipse traversed the continental U.S. from Oregon to South Carolina (Fig. 1) (https://eclipse2017.nasa.gov/eclipse-maps). Although the path of totality covered a small swath about 100 kilometers wide, the penumbra extended from the tropics to all of North America up to the arctic polar limit, about 6400 km in diameter. Thus, the solar eclipse can cause large reductions in both temporally averaged surface broadband shortwave (SW) flux at a given site along the totality path and spatially averaged global surface SW radiation budget at a given time during the eclipse. The eclipse-induced surface SW flux reduction can lead to a decrease in sensible heat flux and associated changes in wind speed (e.g. Turner et al., 2018). As some geo-engineering ideas suggest the blocking or reflecting of solar radiation back to space, the testing of our quantitative understanding of solar radiation in obscured situations is critically important (National Research Council, 2015). Thus, quantifying and understanding the changes of the surface SW irradiances during a solar eclipse is important in this natural experiment.

Several ground-based radiation experiments and modeling activities have been carried out for understanding radiation in solar eclipse conditions in the past. Sharp et al. (1971) reported that the sky light may be considered as attenuated sunlight up to at least 99.8% obscuration and the effect of multiple scattering from outside the umbral region dominates the sky brightness close to and during totality (e.g. Mikhalev et al.,1999; Zerefos et al., 2000). Shaw et al. (1978) developed a model to compute sky radiance during a total solar eclipse by including first- and second-order scattering processes that would compute the diffused light scattered into the umbra. Emde and Mayer (2007) performed a full 3D radiative transfer model exercise to simulate surface spectral solar radiance and irradiance change for cloudless atmosphere during a total eclipse on 29 March 2006, providing a benchmark for studying radiative transfer under solar eclipse conditions.

During the 21 August 2017 solar eclipse, Bernhard and Petkov (2019) made surface spectral solar irradiance observations and performed 3D radiative transfer simulations; Ockenfuβ et al (2019) further simulated 3D radiative transfer in more detail for understanding the impact of surface spectral albedo, ozone vertical distribution and surrounding mountains on surface spectral irradiance observed by Bernhard and Petkov (2019).

Estimating the impact of an eclipse on surface SW flux is a challenging task. Though one may observe the variation of SW flux variations during an eclipse from ground-based radiometers, it is almost impossible to obtain the observations for the same atmospheric conditions but without a solar eclipse because the atmosphere is often cloudy and cloud properties change rapidly from the beginning to the end of a solar eclipse. In the past, most observations and radiative transfer modeling studies for solar eclipse conditions focused on spectral irradiance change during a solar eclipse. Although there were some surface SW irradiance observations (e.g. Koepke et al., 2001; Calamas et al., 2018), there is a lack of the quantification of the solar eclipse's impact on the surface SW flux mainly because of the complicating presence of clouds.

Clouds cover a large part of the Earth. The average global cloud cover is about 68% for cloud optical depth > 0.1 and about 56% for cloud optical depth larger > 2. Locations on the totality path are often covered by clouds. Quantifying the impact of an eclipse on time-averaged local surface broadband SW flux in cloudy atmospheric conditions and estimating the influence





on global surface flux reduction by the Moon's shadow from ground-based observations are the main objectives of this

study.

This ground-based measurement paper complements that of Herman et al.'s (2018) paper on the reduction of reflected spectral radiance based on DSCOVR/EPIC top of the atmosphere (TOA) observations. In this study, we combined radiometer observations with a radiative transfer model to estimate the impact of the solar eclipse on the temporally averaged SW flux at Casper, Wyoming and Columbia, Missouri. We further estimated the reduction of the global average surface SW

radiation when the totality occurred at the two sites. Since both sites were covered by clouds, this study focuses on understanding the role of cloud in irradiance reduction during the eclipse.

In Section 2 of this paper, we describe the ground-based solar radiation experiments. Section 3 describes the radiative transfer modeling experiment. The methodology is presented in Section 4. The results are presented in Section 5 followed by the summary in Section 6.

## 75    2 Ground-based Observation Experiments

Two ground sites were carefully selected from the totality path of the 21 August 2017 eclipse. They were Casper, Wyoming (at 42°50.2′ N, 106°19.4′ W) and Columbia, Missouri (at 38°57.1′N, 92°20.1′ W); both were near the center of the path of totality and experienced a nearly overhead total solar eclipse (local time solar time 10:38am in Casper and 12:04pm in Columbia) (see Fig. 1 and Table 1 for detailed information). These two sites are separated by a distance of about 1200 km, a

typical synoptic scale, such that the weather at these sites can be quite different, allowing us to study the eclipse-induced surface SW changes under different atmospheric conditions.

The ground-based instruments include a thermal-dome-effect-corrected (TDE) pyranometer (Ji and Tsay, 2010), a standard Pandora spectrometer instrument system (PSI) for 280-520 nm wavelength range (Herman et al., 2009) and an extended-range PSI (PSI-ER) for 280-820 nm wavelength range (Jeong et al., 2018) at both sites. The pyranometer is a broadband

radiometer that measures solar radiation reaching Earth's surface with wavelengths approximately from 295 nm to 2800 nm. Ji and Tsay (2000) found that the fused silica dome's thermal effect on the pyranometer can introduce an error a few W m$^{-2}$ to over tens of W m$^{-2}$ depending on the temperature difference between its thermopile and glass-filter domes. Ji et al. (2011) developed a novel nonintrusive method to correct the pyranometer's TDE and demonstrated a high level of consistency with NIST-traceable light source maintained in a Class 10,000 clean room at the NASA Goddard Calibration Facility. Reported

accuracy of this light source for the calibration is better than 1%. The pyranometer-observed surface broadband SW flux without TDE correction at the totality is about -13 W m$^{-2}$ and -5 W m$^{-2}$ at the Casper and Columbia site, respectively. However, these unrealistic negative biases during the totality are improved with the TDE correction (the SW fluxes are 5 W m$^{-2}$ at Casper and -3 W m$^{-2}$ at Columbia). Note that according to the results of Emde and Mayer (2007), surface spectral irradiance (therefore broadband SW flux) for eclipse conditions is four orders of magnitude smaller than its counterpart for a

non-eclipse condition. Therefore, theoretical broadband SW fluxes at these sites are less than about 0.1 W m$^{-2}$. Although





these small offsets of about ± 5 W m$^{-2}$ are still greater than those of typical nighttime biases with the TDE correction (*e.g.* Tsay et al., 2019), they can be attributed to the abrupt variation of SW fluxes during the eclipse coupled with radiometric performance of the sensors and calibration uncertainties. We subtract the offset from the observations such that the surface SW flux is zero at the totality for both Casper and Columbia sites.

Both PSI and PSI-ER contain a small Avantes low stray light spectrometer. The optical head consists of a collimator and filter wheels giving rise to a 2.2° field of view (FOV) for direct-Sun measurements. The PSI is capable of obtaining $NO_2$ and ozone total column amounts (for details, see Herman et al., 2009, 2015). The PSI-ER has the capability to retrieve aerosol and cloud optical depths within the given wavelength range (Jeong et al., 2018). Note that cloud optical depth is usually much larger than aerosol optical depth. As cloud optical depth increases, the direct sunlight decreases exponentially, leaving

a very small signal for an instrument to detect. We used only data with a signal-to-noise-ratio (SNR) greater than 10.

The current PSI does not have an operational scheme for water vapor retrieval. The precipitable water vapor amount observations from the nearest AERONET stations (see Table 2) were used in radiative transfer computations for the Columbia and Casper sites, respectively.

## 3 Radiative Transfer Model and Model Inputs

### 3.1 The Model

The radiative transfer model used is a fast plane-parallel broadband model for both solar shortwave and terrestrial longwave irradiances originally developed by Fu and Liou (1992) and subsequently modified by the SARB (Surface and Atmospheric Radiation Budget) team at NASA's Langley Research Center (Kato et al., 2005; Rose et al., 2006). The SW portion of the model used in this study is a delta-four-stream radiative transfer code with 18 spectral bands from 0.175 μm to 4.0 μm. The

model accounts for gaseous absorption by $O_3$, $H_2O$, $O_2$, $CO_2$ and $CH_4$, molecular scattering, aerosol and cloud absorption and scattering. We also used the SBDART (Santa Barbara DISORT Atmospheric Radiative Transfer) model (Ricchiazzi et al., 1998) to simulate the surface spectral flux for TOA spectral solar irradiance for both normal and eclipse conditions to understand the role of clouds on transmitted spectral and total shortwave flux.

The assumption of constant collimated incident solar intensity in the 1D model is invalid for the umbra and near the totality

region because the surface diffuse component, which depends on the 2D distribution of the TOA incident solar irradiance, dominates under those conditions. Emde and Mayer (2007) performed a rigorous analysis to quantify 1D errors in diffuse spectral radiance and irradiance as a function of the time from the center of the totality. We used their results for spectral irradiance at 500 nm as a surrogate for estimating the error in broadband shortwave irradiance because the solar spectrum peaks near 500 nm.

For a plane-parallel clear atmosphere, one can show that the surface diffuse flux is about 10% of the direct component at 500 nm for solar zenith angles (SZA) from 0° to 40°. Thus, a 10% 1D error in the diffuse component at time 150 seconds (about





km) from the center of the totality will lead to about 1% error in total surface SW flux estimate. Further away from the totality, the direct component gradually dominates and the 1D error in the diffuse flux decreases quickly with distance (see Fig. 14 in Emde and Mayer, 2007), resulting in an even faster decrease of the 1D error in total surface SW flux. Thus, the

error in the average shortwave irradiance from the 1D model is negligible.

Additionally, cloud inhomogeneity can introduce large uncertainties in 1D radiative transfer models, and is a major obstacle for computing radiative flux for solar eclipse conditions (e.g. Koepke et al., 2001). We will discuss this issue in Section 4.

### 3.2 Model Inputs

### 3.2.1 TOA Spectral Solar Irradiance During the Eclipse

The change of TOA spectral solar irradiance is essential for modeling solar radiation transfer during an eclipse. For normal conditions, the extraterrestrial solar irradiance at each wavelength is given as an average over the whole solar disk. For eclipse conditions one needs to integrate the limb darkening function weighted spectral irradiance for the non-obscured part of the Sun to obtain the TOA spectral solar irradiance. Here we adopted the analytical expression by Koepke et al. (2001) to compute the spectral solar irradiance emitted from the non-obscured solar disk (or reduced brightness) as a function of the

distance between the centers of the disks of the Moon and the Sun with the limb darkening function from Neckel (2005).

The astronomical aspect of solar eclipse is well understood and the geometry of the problem can be calculated with high accuracy (e.g. Espenak and Anderson, 2004). The parameters for 21 August 2017 eclipse (Table 1) are calculated with an online calculator provided by the Astronomical Applications Department of the US Naval Observatory (USNO) at http://aa.usno.navy.mil/data/docs/Eclipse2017.php. We followed the definition of the distance between the center of the

disks of the Moon and the Sun, normalized by the sum of the radii of Moon and Sun in Koepke et al. (2001). For computing the reduced brightness as a function of time for the two sites for the entire course of eclipse event, we also used the fact that the value of distance is linearly correlated to time (e.g. Koepke et al., 2001).

### 3.2.2 Atmospheric and Surface Properties

The standard mid-latitude atmosphere is used to describe the temperature, pressure, and trace gas profiles. Two major

absorbing gases for shortwave radiation, ozone and water vapor, are based on observations; other less important trace gases are kept at constant levels. Column ozone amount observations from the EPIC at 15:44:50 UTC before the eclipse are used for the Casper site. The column ozone from PSI before the eclipse is used for the Columbia site. The precipitable water vapor amounts are from nearby AERONET stations (see Table 2). The ozone and water vapor profiles are scaled to match the observed total column amounts. Aerosol optical depth (AOD) was observed by PSI-ER before the eclipse and the aerosol

type is assumed to be continental aerosol with scale height of 3 km. All trace gases and AODs are assumed constant in radiative transfer calculations.



PSI-ER was operating continuously at both sites to provide optical depth observations. Using Beer's law for a constant TOA solar irradiance ($I_0$), one can obtain apparent optical depth from Eq. (1)

$$I(t) = I_0 e^{-\frac{\tau_{app}(t)}{\mu_0(t)}} , \tag{1}$$

where $I(t)$, $\tau_{app}(t)$, and $\mu_0(t)$ are the PSI-ER observed irradiance, the apparent optical depth, and cosine of solar zenith angle at time $t$, respectively. Without considering the decrease of TOA solar irradiance during solar eclipse, Eq. (1) will lead to a much larger apparent optical depth than it should be. Thus, one has to use the reduced TOA solar irradiance that accounts for limb darkening effects to derive the true optical depth in Eq. (2)

$$I(t) = I_{0,eclipse}(t) e^{-\frac{\tau(t)}{\mu_0(t)}} , \tag{2}$$


where $I_{0,eclipse}(t)$ and $\tau(t)$ are the true TOA solar irradiance and optical depth. From Eqs (1) and (2) one can derive the true optical depth as a function of apparent optical depth and the ratio of solar irradiances with and without solar eclipse in Eq. (3)

$$\tau(t) = \tau_{app}(t) + \mu_0(t) \ln \left( \frac{I_{0,eclipse}(t)}{I_0} \right) . \tag{3}$$

Subtracting the molecular scattering optical depth and aerosol optical depth from the total optical depth, we derive true cloud optical depth. The apparent and true total optical depths are presented in Fig. 2.

From the ground, the authors at the site observed that the atmosphere over the Casper site was mostly clear with some thin cirrus clouds. The visible images from GOES-16 satellite (Schmit et al., 2005) captured the eclipse and showed a fraction of cirrus cloud near the Casper site before, during, and after the eclipse. Examples of two GOES-16 images are presented in 175     Figure 3a,b. The GOES-16 images and Sun-pointing PSI-observed cloud optical depth suggest the presence of thin cirrus clouds not shading the direct solar beam for some time before and during a large part of the eclipse, with some thin cirrus fragments passing intermittently through the FOV of the PSI. The photo taken near the totality captured a moment of the sky when the direct solar beam was shaded by a thin cirrus cloud (Fig. 3(c)). Terra satellite passed over at 17:45 UTC, the time of totality at the Casper site. The average cloud top pressure from Moderate-Resolution Imaging Spectroradiometer 180     (MODIS) thermal channel observations was approximately 327 mb.

As observed by the authors at the site, the sky over the Columbia site was covered by cirrus clouds above some scattered low and mid-level cumulus clouds (Fig. 3(f)). The radiosonde relative humidity profile from the nearest station before the eclipse suggests a multi-layer cloud system with cloud tops near 200, 400, and 650 mb (Fig. 4). The GOES-16 satellite thermal infrared images show that the Columbia site was always covered by high-level clouds as indicated by very low brightness 185     temperature (about -20°C to -40°C) (Fig. 3(d), (e)). The Suomi National Polar-orbiting Partnership (Suomi NPP) satellite (Hillger et al., 2013) overpassed the Columbia site at 18:30 UTC when the site was in partial eclipse. The average cloud-top-



height from Visible Infrared Imaging Radiometer Suite (VIIRS) thermal infrared retrieval around the Columbia site was about 230 mb.

Because the clouds are optically thin during most of the eclipse for both sites except the two large spikes near 17.7 and 18.5

UTC at the Columbia site (Fig. 2), we assumed one-layer cirrus cloud between 200 and 400 mb with effective diameter of 60 μm in the Fu and Liou (1992) radiation code for computing the surface SW flux. We will compare the model results with observations and discuss the error cloud inhomogeneity not accounted for in the 1D model in Section 5.

Surface spectral albedo is based on the monthly average value from MODIS product and International Geosphere-Biosphere Programme (IGBP) albedo. We combine MODIS surface spectral albedo at 7 bands from 0.47μm to 2.13μm (Schaaf and

Wang, 2015) and albedo from IGBP to get spectral albedo for the 18 bands in the Fu&Liou model. By using these estimates of atmospheric composition and radiative algorithms, we are able to estimate the amount of radiation reaching the Earth's surface during an eclipse.

## 4 Methods

### 4.1 Deriving Surface Irradiance for Non-eclipse Conditions

Koepke et al. (2001) estimated the photolysis frequencies for non-eclipse conditions using the observed photolysis frequencies during an eclipse divided by the normalized radiance. This method can be applied to estimate surface spectral radiance and irradiance for non-eclipse conditions. In this section, we will show that the surface broadband SW flux for non-eclipse conditions can be estimated from ground-based pyronomter observed flux during the eclipse.

The surface broadband SW flux may be expressed as

$$F = \int I(\lambda) T(\lambda) d\lambda \, , \qquad (4)$$

where $I(\lambda)$ and $T(\lambda)$ are incident TOA spectral solar irradiance and atmospheric transmittance at wavelength λ, respectively. We demonstrate the effect of an eclipse on the distribution of the TOA spectral solar irradiance and influence of clouds on the transmittance in Fig. 5. Here we define the total normalized spectral irradiance as

$$I_{norm}(\lambda) = \frac{\int I_{non-eclipse}(\lambda) d\lambda}{\int I_{eclipse}(\lambda) d\lambda} I_{eclipse}(\lambda) \, , \qquad (5)$$

where $I_{eclipse}(\lambda)$ and $I_{non-eclipse}(\lambda)$ are TOA spectral solar irradiance at wavelength $\lambda$ for eclipse and non-eclipse conditions; the spectrally integrated irradiance of $I_{norm}(\lambda)$ is always equal to the TOA total solar irradiance for non-eclipse conditions. Fig. 5(a) shows that there is a red-shift in TOA spectral solar irradiance as obscuration increases since the limb darkening has a much stronger effect at shorter wavelengths (e.g. Koepke et al., 2001). The peak of the spectral irradiance shifts from 0.45μm for non-eclipse condition to 0.50μm and 0.58μm for 90% and 99% obscuration of solar disk,

respectively. $I_{norm}(\lambda)$ is also called red-shift spectral solar irradiance. Note the true TOA irradiance decreases by one order





of magnitude from normal condition to 90% obscuration and from 90% to 99% of obscuration during eclipse (see the inset of Fig. 5(a)).

Clouds play a unique role in modifying spectral solar irradiance reaching the surface. We used the SBDART to compute spectral transmittance as a function of cloud optical depth for different TOA solar spectra. Fig. 5(b) shows that an increase of

cloud optical depth leads to a relatively larger decrease of surface spectral irradiance in near-IR wavelengths compared to near-UV and visible wavelengths. Here we examine the effect of cloud on transmitted flux for red-shift spectral solar irradiance. For the red-shift spectrum, an increase in cloud optical depth leads to a relatively smaller decrease in transmitted surface flux in near-UV and visible wavelengths. There is a relatively larger decrease in near-IR wavelengths compared to the spectrum for the normal conditions simply because of the red-shift in TOA solar spectrum. To some extent, the larger

decrease in near-IR wavelengths compensates for the smaller decrease in visible and near-UV wavelengths, resulting in a decrease in spectrally integrated surface SW flux similar to that for the normal TOA spectral solar irradiance.

Figure 5c shows the change of the spectrally integrated SW flux calculated from the SBDART as a function of cloud optical depth at 0.55 µm for the normal solar spectrum and red-shift spectral solar irradiance associated with different obscuration levels (Fig. 5a), and shows that all curves of surface SW flux are similar in shape. For a given cloud optical depth, there is a

slight larger decrease in surface SW flux for a larger red-shift TOA solar spectrum associated with a larger obscuration. The ratio of surface SW flux for the normal TOA solar spectrum to that for the red-shift solar spectrum is presented in the inset in Fig. 5(c). It is clear that the flux ratio is not very sensitive to cloud optical depth and the ratios are slightly larger than unity. Note that one needs to multiply a scale factor of $\int I_{non-eclipse}(\lambda)\,d\lambda / \int I_{eclipse}(\lambda)\,d\lambda$ to the ratios in the inset in Fig. 5(c) to obtain the true non-eclipse-to-eclipse surface SW flux ratio. Thus, the surface SW flux ratio depends on the obscuration of

the eclipse and is not very sensitive to cloud optical depth.

Figure 5(d) shows the time series of the modelled non-eclipse-to-eclipse surface SW flux ratio for clear atmosphere and cloudy atmosphere with cloud optical depth of 2 for the Columbia site. The difference between the two ratios is less than 1%. The difference increases slightly with cloud optical depth. For a cloud optical depth of 10, the difference is close to 4% near to totality at 99% obscuration.

In this study, we assume that the non-eclipse-to-eclipse surface SW flux ratio for realistic 3D cloudy atmospheric conditions is approximately equal to the 1D model computed flux ratio for clear atmospheric conditions, i.e.,

$$\frac{F_{non-eclipse}(t)}{F_{eclipse}(t)} \approx \frac{F_{non-eclipse,model}(t)}{F_{eclipse,model}(t)}, \qquad\qquad (6.1)$$

where $F_{eclipse}(t)$ and $F_{non-eclipse}(t)$ are surface SW fluxes observed by pyronometer and what it would be observed without solar eclipse, $F_{eclipse,model}(t)$ and $F_{non-eclipse,model}(t)$ are the counterparts from a 1D model for clear conditions at

time $t$, respectively. Thus, the surface SW flux for non-eclipse conditions can be estimated as

$$F_{non-eclipse}(t) \approx \frac{F_{non-eclipse,model}(t)}{F_{eclipse,model}(t)} F_{eclipse}(t) . \qquad\qquad (6.2)$$



## 4.2 Estimating the Impact of the Eclipse on Global Average Surface Broadband SW Flux from Ground-based Observations

In addition to estimating the impact of the eclipse on time average flux at two local sites, we also estimate its influence on the global average surface SW radiation budget. During a solar eclipse, the Moon casts a shadow that extends to an area greater than 3000 km in radius, significantly reducing the global average surface SW radiation budget. Estimating the impact of a solar eclipse on the global shortwave radiation budget from local observations is a major goal of this research. First, we present a method for computing the change of the global averaged surface SW flux from spatially averaged observations. Then we extend these results to global average irradiance reduction.

First, the global average surface SW flux for eclipse condition is the area weighted flux inside and outside of the Moon's shadow; it can be written as

$$F_1 = \frac{(\pi R_e{}^2 - \pi r_0^2)F' + \pi r_0^2 F_{eclipse}}{\pi R_e{}^2}, \tag{7.1}$$

where $R_e$ is Earth's radius, $r_0$ is the radius of penumbral shadow projected on Earth cross-section perpendicular to Sun-Earth line (the outermost circle in Fig. 6), $F'$ is the average flux outside of the Moon's shadow, and $F_{eclipse}$ is the average flux in

the Moon's shadow. Similarly, for non-eclipse condition, the global average surface SW flux is

$$F_2 = \frac{(\pi R_e{}^2 - \pi r_0^2)F' + \pi r_0^2 F_{non-eclipse}}{\pi R_e{}^2}, \tag{7.2}$$

where $F_{non-eclipse}$ is the average surface SW flux for the Moon's shadow area as if the eclipse were not present.

The eclipse-induced relative reduction of surface SW flux to the global average value is

$$\Delta F = \frac{F_1 - F_2}{F_2}, \tag{8.1}$$

or

$$\Delta F = \frac{F_{eclipse} - F_{non-eclipse}}{F_2} \frac{r_0^2}{R_e^2}, \tag{8.2}$$

where $F_2$ is the global average surface SW flux for non-eclipse conditions. This value may be estimated by multiplying the TOA average total solar irradiance of 1360.8 W m$^{-2}$ (Kopp and Lean, 2011) (with adjustment for the Sun-Earth distance) by the global average transmittance of 0.55 (Trenberth et al., 2009), $R_e = 6378\ km$, and $r_0 = 3430\ km$ calculated using the

geometric information (i.e. Sun-Earth distance, Moon-Earth distance, radii of the Sun and Moon) from the United States Naval Observatory (USNO) website (http://aa.usno.navy.mil/data/docs/geocentric.php). Thus, one needs to know the average surface SW flux for both eclipse and non-eclipse conditions to compute the fractional reduction in global average surface SW flux.



We next show that the temporally resolved downward shortwave flux from the pyranometers may be used to estimate the

spatial average flux in the penumbra, mainly because the ground sites are in the path of the total eclipse; therefore, the

instruments were able to sample the full course of the eclipse.

First, we demonstrate this for an ideal scenario with horizontal homogeneous atmosphere and constant surface albedo. Fig. 6

shows the DSCOVR/EPIC image acquired at 18:14:50 UTC when the Columbia site was experiencing the totality. The

average surface SW flux in the penumbra may be estimated by averaging observations $(F(X_1), F(X_2), ..., F(X_n))$ from a

series of $n$ pyranometers uniformly distributed along the totality path (i.e. $F_{eclipse} = \frac{1}{n}\sum_{i=1}^{n} F(X_i)$). At the Columbia site, the

pyranometer observed a temporal variation of downward flux with uniform increments of time (i.e. $F(t_1), F(t_2), ...., F(t_n)$).

At time $t_1$ when the eclipse started, the surface radiometer sampled the downward flux $F(t_1)$, which would be

approximately the same as the observed flux at the eastern edge (i.e. $F(X_1)$) of the penumbra when Columbia was

experiencing totality. Similarly, the pyranometer observed the surface SW flux at time $t_i$, which would be the same as that

from the pyranometer at $X_i$ in the totality path (the white dashed line in Fig. 6) with the same phase of obscuration (i.e.

$F(X_i) = F(t_i)$). Thus, the temporal average of the observed surface SW flux from $n$ time step from a local site is

approximately equal to the spatial average of the surface SW flux observed from a series of $n$ radiometers.

To estimate the surface SW flux reduction in the whole area of penumbra, one needs to calculate the average flux in the

Moon's shadow. For the assumed homogeneous atmosphere and surface properties, the surface SW flux depends only on the

radius from the totality, and the reduction of the global average flux can be written as

$$\Delta F = \frac{\iint (F_{eclipse}(r) - F_{non-eclipse}(r))r d\varphi dr}{\pi R_e^2 F_2},$$  (8.3a)

where the distance $r$ is the distance from the totality and $\varphi$ is the azimuth angle. Assuming the fluxes are independent of

azimuth angle, Eq. (8.3a) becomes

$$\Delta F = \frac{\int_0^{r_0} (F_{eclipse}(r) - F_{non-eclipse}(r))2\pi r dr}{\pi R_e^2 F_2},$$  (8.3b)

where the distance $r$ is estimated from the linear relation between $r$ and $t$ such that $r = 0$ at the totality and $r = r_0$ at the

beginning and end of the partial eclipse, and $F_{eclipse}(r = X_i) = F_{eclipse}(t_i)$ and $F_{non-eclipse}(r)$ is derived from

$F_{non-eclipse}(r)$ (Eq. 6.2).

From the observed surface SW flux ($F_{eclipse}$), one can estimate the surface SW flux for non-eclipse conditions ($F_{non-eclipse}$)

at each time step as described in Section 4.1 and further to estimate eclipse-induced reduction on global average surface SW

budget (Eq. (8.3)).

We emphasize that the temporal average value from one location represents the spatial average for similar atmosphere and

surface conditions in the penumbra. The results from the Casper site represent mostly clear atmospheric condition. With





more cloud cover over the Columbia site, the estimated shortwave irradiance change is closer to realistic atmospheric condition as described later.

## 5 Results

Figure 7 shows both the observed surface SW flux and derived counterpart for non-eclipse condition from Eq. (6.2) for both sites. It also shows the modelled surface SW fluxes, including the clear sky flux for both eclipse and non-eclipse scenarios and the flux for the one-layer cirrus with variable cloud optical depth for non-eclipse conditions.

For the Casper site (Fig. 7(a)), in the first period from 16 to 18.2 UTC before and during a large part of the eclipse, the observed surface SW flux varies rather smoothly with time, similar in behaviour to that for modelled clear sky flux, except for a few tiny dips which is likely due to fragments of thin cirrus passing through the FOV of PSI as indicated by small spikes in cloud optical depth observations (Fig. 2). From 16 to 16.7 UTC, the observed flux exceeds the modelled one for clear atmospheric conditions by more than 20 W m$^{-2}$ and by a much smaller amount as time proceeds after 16.7 UTC. This enhancement can be explained by the presence of some thin cirrus clouds not shading the direct solar beam in this time period. Thin cirrus clouds not shading the direct solar beam have no impact on the direct component of surface SW flux but increases the downward diffuse radiation, resulting in an increase in total surface SW flux compared to clear atmospheric conditions. The cirrus clouds induced surface SW flux enhancement decreases with time towards the totality as the TOA brightness decreases. In the second time period from 18.2 to 19.2 UTC, the dips in the observed flux are much larger and last longer in time compared to the dips in the first period. This is associated with the nature of the clouds that shade the direct solar beam as indicated by the cloud optical depth observations (see Fig. 2).

For non-eclipse conditions, the cirrus clouds induced enhancement and the downward dips in the estimated surface SW flux are more pronounced compared to the eclipse scenario. In the first time period (16-18.2 UTC), the estimated surface SW flux exceeds that for clear atmospheric conditions by about 20 W m$^{-2}$ in the beginning of the time series to about 100 W m$^{-2}$ around 17.3-17.5 UTC, much larger than the counterpart for eclipse conditions. The dips in the second period (18.2-19.2 UTC) are evidently larger than their counterparts for the eclipse conditions. The magnitude of the dips in the estimated surface flux is closely related to the observed cloud optical depth.

In the first time period (16-18.2 UTC), the modelled surface SW flux (red curve) is close to the clear sky flux (dashed blue) because of the small cloud optical depth and underestimates the surface flux accordingly. However, the model overestimates the surface flux (green curve) in the second period (18.2-19.2 UTC). For a given observed cloud optical depth, we expect the model to provide accurate direct surface SW flux. The discrepancy between the model and observations comes from the difference in the diffuse component. The underestimate in the first time period is due to the fact that the 1D model does not consider the cirrus cloud induced enhancement by the diffuse radiation, which is a 3D effect. The overestimate in the second time period (red curve vs. green one) is because the 1D horizontally extended clouds produce more downward diffuse SW flux than the real cirrus clouds that cover only a fraction of the atmosphere as shown in GOES-16 images (see Fig. 3(a),(b)).





Using the observed and derived surface SW flux for eclipse and non-eclipse conditions, we estimated the average reduction of the local surface SW flux about 379 W m$^{-2}$ or 50%, which corresponds to 8% reduction in the global surface SW radiation when the Moon's shadow was centered at Casper.

Similarly, the variations of the observed surface SW flux at the Columbia site (Fig. 7(b)) can be understood by comparing it with the modelled flux for clear atmosphere during the eclipse. From 16.6 UTC to 17.1 UTC, the observed flux decreases

from 800 to 460 W m$^{-2}$; which is about a 340 W m$^{-2}$ decrease compared to a decrease of about 60 W m$^{-2}$ for clear atmospheric condition (blue curve). This much larger decrease in the observations is primarily due to the increase of cloud optical depth during this time period (see Figs. 2(b),8(b)). From 17.1 to 17.4 UTC, there is a slight increase in the observed surface SW flux compared to a continuous decrease of the SW flux for the clear atmospheric conditions. The slight increase in the observed surface SW flux is the combination of the decrease of the cloud optical depth and the decrease in the TOA

brightness. Thus, the observed cloud optical depth combined with the TOA brightness can be used to interpret the main features of observed surface SW flux variations. There are time periods when observations exceed the values for clear atmosphere by nearly 50 W m$^{-2}$ in 18.65-18.8 UTC and 80-100 W m$^{-2}$ in 19.2-19.6 UTC.

For non-eclipse conditions, the cloud effects of reducing and enhancing the surface flux are more pronounced compared to the eclipse conditions similar to the results for the Casper site. The derived non-eclipse flux exceeds the value for clear

atmospheric conditions by 150 W m$^{-2}$ (18%) at 18.65-18.8 UTC and near 100 W m$^{-2}$ (12%) at the end of the eclipse in 19.2-19.6 UTC. Koepke et al. (2001) suggested that when the direct solar beam is not shaded by a cloud, the additional reflection of solar radiation from vertically extended clouds can increase the incoming surface radiation by up to 25% above the corresponding cloud-free values. Thus, it is not surprising to see a large enhancement of surface SW flux in a system of cumulus clouds under optically thin cirrus clouds.

In non-eclipse conditions, we found that the 1D model (red curve) overestimates the surface flux (green curve) for most situations. Again, the cloud inhomogeneity is the main cause of the overestimation. The low and mid-level cumulus clouds that are not accounted for with 1D model reflect the diffuse radiation scattered by cirrus clouds above them; as a result, a smaller amount of diffuse radiation reaches the detector, thus a smaller total SW flux is measured compared to a 1D model. Evidently, a 1D model is unable to simulate the enhancement induced by cloud side reflection.

From the observed surface SW flux and estimated flux for non-eclipse conditions, we estimated the average reduction of the *local* average surface SW flux as about 329 W m$^{-2}$ or 46%, corresponding to 7% reduction in the *global* average surface SW flux when Moon's shadow was centered at Columbia.

For understanding the role of clouds in eclipse-induced flux reduction we modelled the surface SW flux for different cloud optical depth. Fig. 8 shows that the increase of cloud optical depth leads to a decrease in surface flux for both non-eclipse

and eclipse conditions. However, at a given time during the eclipse, the rate of decrease of surface flux to the increase of cloud optical depth for the eclipse (difference between solid curves) is smaller than the rate for non-eclipse conditions (difference between dashed curve). This is primarily due to a smaller TOA reduced brightness for eclipse conditions.





Figure 9 shows flux difference (i.e. $F_{non-eclipse}(t) - F_{eclipse}(t)$) for different cloud optical depth. It is evident that the flux difference is largest for clear atmospheric conditions; and the difference decreases with the increase of cloud optical depth.

Thus, the eclipse has a smaller impact on surface flux under cloudy compared to clear atmospheric conditions; the impact decreases with the increase of cloud optical depth.

Figures 8 and 9 show that both the time-averaged surface flux for non-eclipse conditions (e.g. the area under the dashed curve in Fig. 8) and the average flux reduction (e.g. the area under each curve in Fig. 9) decrease with cloud optical depth; the ratio of the two does not vary much with cloud optical depth. In fact, Fig. 10 (blue curves) shows that the relative

reduction of the local surface flux is not very sensitive to cloud optical depth, remaining around 45% at Casper and a slightly larger value at Columbia.

The reduction of global SW radiation relative to climatology of surface flux ($F_2$ in Eq. (8.2)) depends on the average flux difference between non-eclipse and eclipse conditions in the Moon's shadow area ($F_{non-eclipse}$ and $F_{non-eclipse}$ in Eq. (8.2)). This flux difference is proportional to the area under each curve in Fig. 9, which always decreases with cloud optical depth.

Thus, the relative reduction of global surface radiation, calculated rigorously using Eq. (8.3), decreases with the cloud optical depth in the Moon's shadow (black curves in Fig. 10).

Figure 10 also shows that, for a given cloud optical depth, the reduction of the average surface SW flux for the Columbia site is larger than for the Casper site. This difference can also be seen from Fig. 9. These differences are mainly due to a smaller SZA at Columbia compared to Casper (see Table 1). The cosine of SZA for the Columbia site is about 10% larger than that

for the Casper site; thus, the average TOA incident solar irradiance for the Columbia site is also about 10% larger than that for the Casper site. For the same optical depth, there is a larger surface SW flux at Columbia site compared to the Casper one for non-eclipse conditions; therefore, the impact of the eclipse on surface flux at the Columbia site is larger than that at the Casper one.

At Casper, the observation-based relative reduction of the *local* surface SW flux (50%) is significantly larger than the 1D

modelled prediction (45%); however, the relative reduction of *global* flux of (8%) is close to the modelled value (8.5%) for the average cloud optical depth. At the Columbia site, the observation-based the relative local reduction of the *local* surface SW flux (46%) is slightly larger than the model prediction (45%); from the other hand, the relative reduction of the *global* flux (7%) is significantly smaller than to the modelled one (9%). These differences between observations and model simulations are mainly due to cloud inhomogeneity not accounted for in the 1D radiative transfer model.

## 6 Summary


We have conducted a ground-based experiment to observe broadband shortwave irradiance at Casper, Wyoming and Columbia, Missouri located in the totality path of the 21 August 2017 solar eclipse. These two sites are separated by a distance about 1200 km and had different atmospheric conditions. Surface shortwave flux measurements with simultaneous



atmospheric observations allow us to study the impact of the solar eclipse on the surface shortwave radiative budget under
different atmospheric conditions.

Radiative transfer calculations show that the non-eclipse-to-eclipse surface SW flux ratio primarily depends on the obscuration of the solar disk during eclipse and slightly depends on cloud optical depth. These results allow us to derive non-eclipse surface SW flux under cloudy atmospheric conditions by multiplying the observed SW flux by the modelled surface SW flux ratio.

We found that at the Casper site, the eclipse led to a decrease of 379 W m$^{-2}$ (50%) in average *local* surface SW flux, and the Moon's shadow caused about a 8% reduction in *global* average surface SW radiation budget when the totality was at Casper; at the Columbia site, the eclipse led to a decrease of 329 W m$^{-2}$ (46%) in average *local* surface SW flux, and the Moon's shadow caused about a 7% reduction in *global* average surface SW radiation budget when the totality was at Columbia.

Clouds play a unique role in modifying the surface flux reduction during an eclipse. The eclipse-induced surface flux
reduction is largest when sky is clear. For opaque clouds, the surface even without eclipse would be already dark to begin with; thus, solar eclipse would have little impact on the surface SW flux. The average flux reduction decreases with the increase of cloud optical depth. However, the relative reduction of *local* surface flux is about 45% and not sensitive to cloud optical depth. The relative reduction of *global* average surface SW flux depends on cloud optical depth in the Moon's shadow and geolocation due to the change in SZA.

We have discussed the 3D effect of clouds on surface radiation. We identified that the presence of cirrus clouds not shading the direct solar beam can significantly enhance the local surface flux; some large flux enhancements may be explained by the reflection of solar radiation by cumulus clouds; some discrepancies between a 1D model and observations may be understood as cloud inhomogeneities not accounted for in a 1D model. The mechanisms of cloud 3D effects on surface radiation enhancement has implications for surface remote sensing research.

*Data availability*. Calibrated pyranometer observed broadband flux and optical depth data are available as a Supplement, the AERONET data are available at https://aeronet.gsfc.nasa.gov, the MODIS and VIIRS data are available at https://earthdata.nasa.gov, and the DSCOVR/EPIC data are available at https://eosweb.larc.nasa.gov/project/dscovr/dscovr_epic_l1b.

*Author contributions*. GW wrote most of the paper and performed most of the analysis with the help from AM. AM, ST, JH,
UJ, and NA participated field experiment to collect radiation measurements. RS helped with instrument management and DW helped with data analysis.

*Competing interests*. The authors declare that they have no conflict of interest.

*Acknowledgments*. This research was supported by NASA's Interdisciplinary Science for Eclipse 2017 program managed by Dr. M. Guhathakurta and partly supported by NASA to the Sun-Climate research.



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

**Figures**

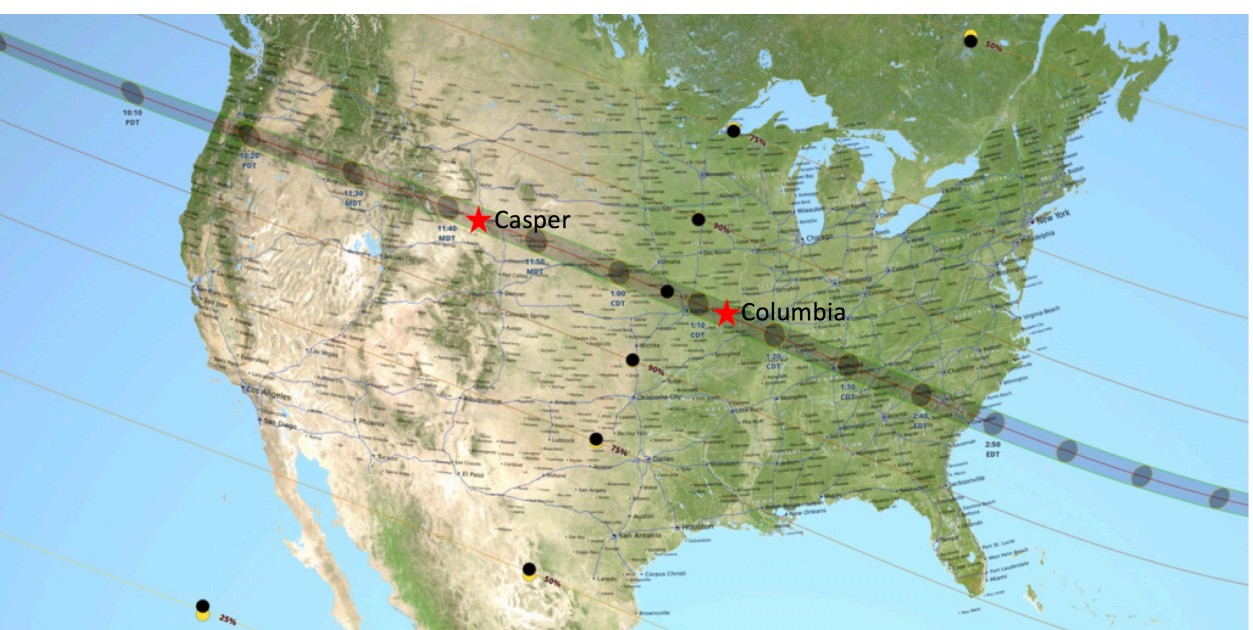

**Figure 1: The eclipse map (from https://eclipse2017.nasa.gov) shows the totality path and obscuration levels on 21 August 2017.**
**Radiometers were deployed to make ground-based observations at Casper, Wyoming and Columbia, Missouri.**


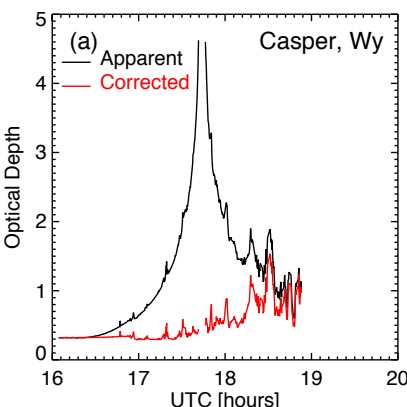
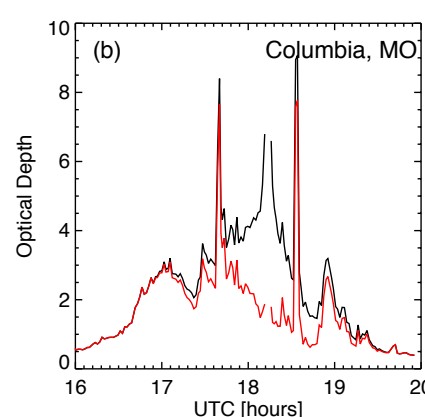

**Figure 2: Apparent (black lines) and corrected (red lines) total optical depths that correspond to radiances observed by Pandora systems at (a) Casper and (b) Columbia during solar eclipse on August 21, 2017.**

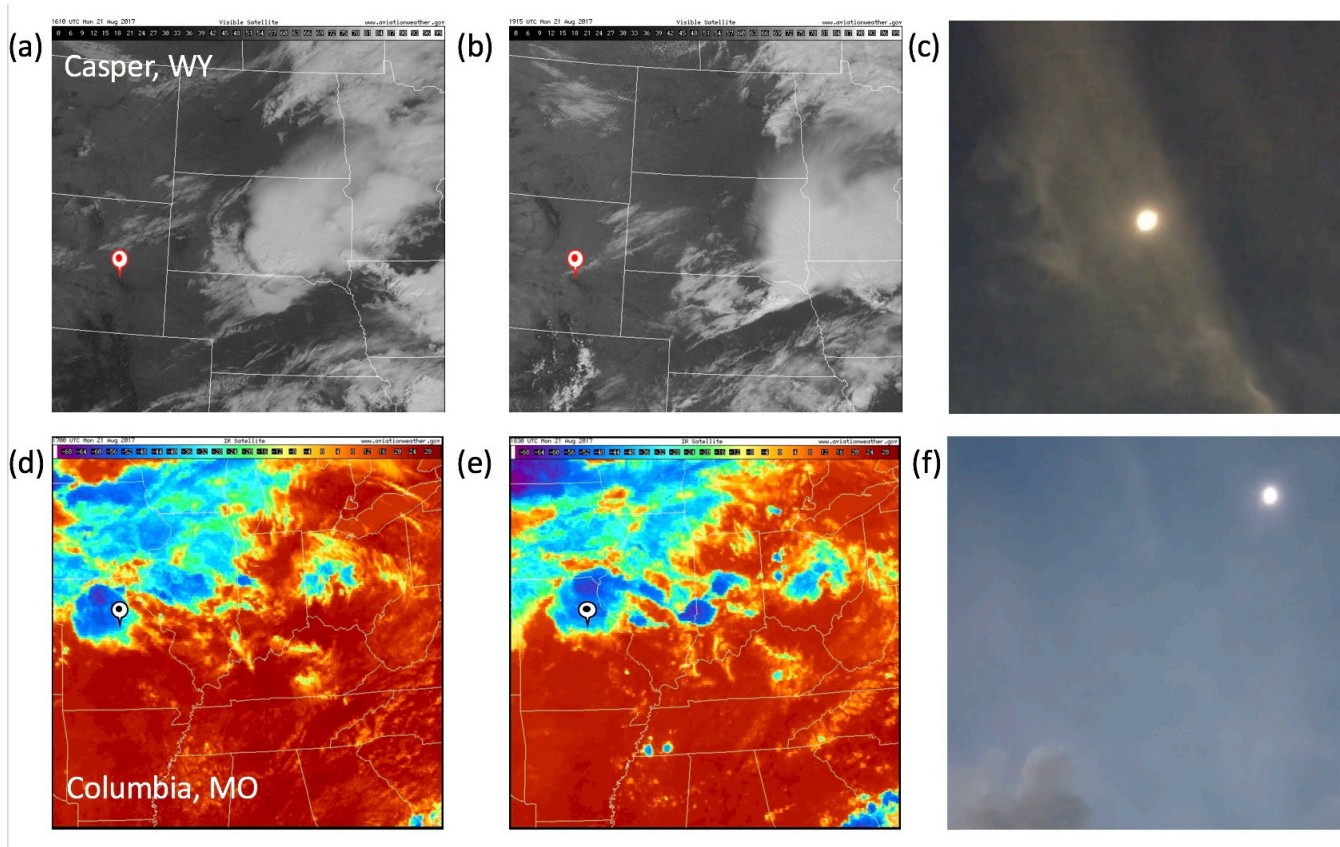

**Figure 3: Top panel for Casper: (a) and (b) are geostationary satellite (GOES-16) visible images at 16:10 UTC and 19:15 UTC, showing thin cirrus clouds over the Casper site indicted by the mark; (c) photo taken near the totality. Lower panel for Columbia: (d) and (e) are the thermal infrared images 17:00 UTC and 18:30 UTC, showing high level clouds over Columbia site indicated by**





the mark; (f) photo taken close to the totality. The satellite images were downloaded from the National Center for Atmospheric Research image archive at http://www2.mmm.ucar.edu/imagearchive/.

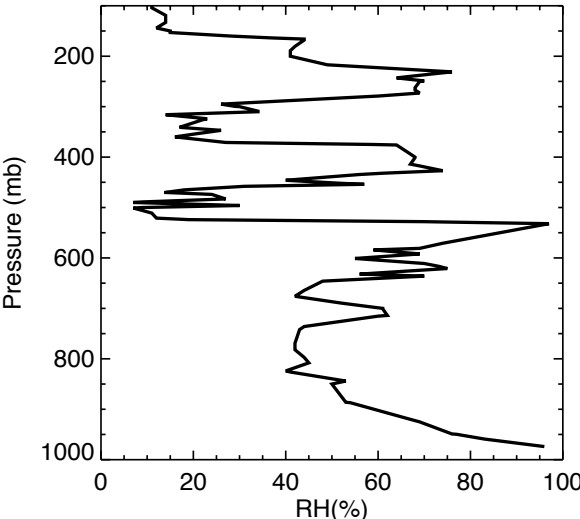


Figure 4: Radiosonde observed vertical profile of relative humidity from nearest station in Springfield, MO (at 37°14' N, 93°24' W) at 12 UTC on 21 August 2017 obtained from http://weather.uwyo.edu/upperair/sounding.html.


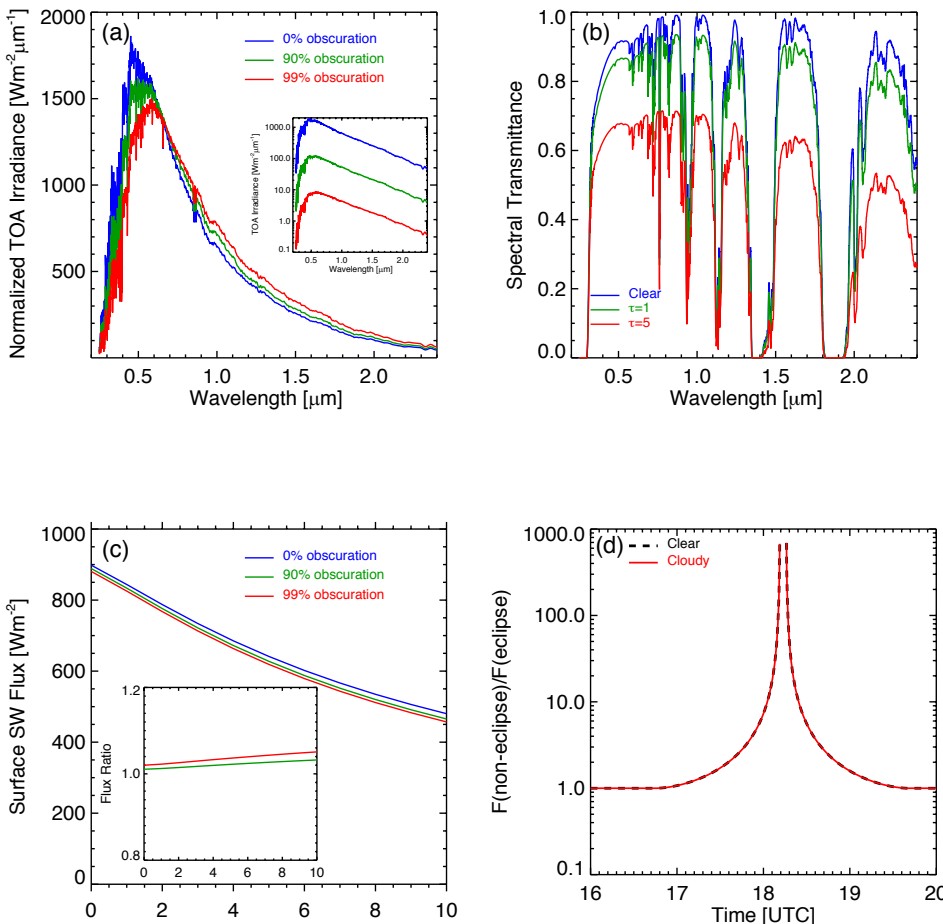

**Figure 5: (a) Normalized TOA spectral solar irradiance such that the spectrally integrated total irradiances equal to that for normal condition (0% obscuration) with the true irradiances shown in the inset. The spectra are peaked at 0.45μm, 0.50μm, and 0.58μm for normal condition (0% obscuration), eclipse conditions with 90% and 99% of obscuration; (b) spectral transmittance for clear and cloudy atmospheres for SZA = 30° calculated from the SBDART; (c) the SBDART modelled surface SW flux as a function of cloud optical depth for different TOA solar spectrum in (a) with the ratio of surface SW flux for normal spectrum to that for different red-shift spectrum in the inset; (d) the Fu&Liou radiation code modelled non-eclipse-to-eclipse surface SW flux ratios for clear atmosphere (dashed black) and cloudy atmosphere with cloud optical depth of 2 (red) from 16 UTC before the eclipse to 18.19 UTC (99% obscuration) and from 18.27 UTC (99% obscuration) to 20 UTC after the eclipse.**



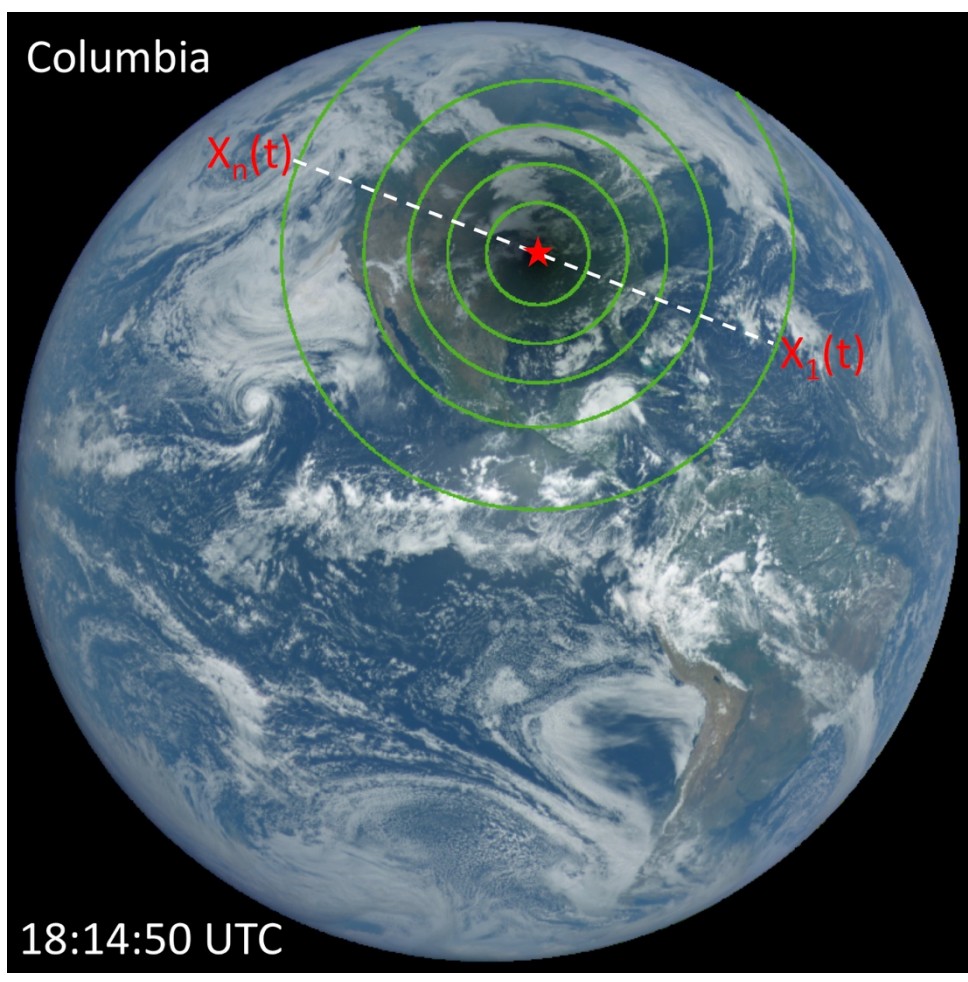

**Figure 6: A sketch illustrating the conversion from temporal to spatial average. The color image has been adjusted from the images on https://epic.gsfc.nasa.gov by increasing the gamma correction (Cescatti, 2007) to bring out the region of totality over Columbia (red star) and surrounding clouds. The green contours show the levels of obscuration from 0% for the outmost circle with decrement of 20% inward. The dashed line illustrates the totality path.**





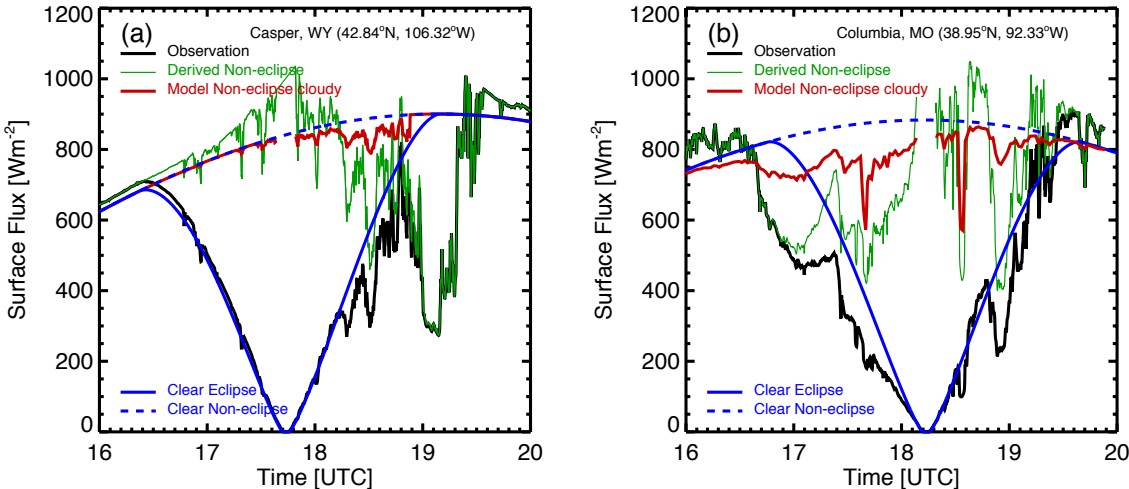

**Figure 7: (a) Casper, (b) Columbia. Observed surface flux (black), derived surface SW flux for non-eclipse conditions (green), surface flux for clear atmospheric condition for eclipse (solid blue) and non-eclipse conditions (dashed blue), the modelled surface flux (red) uses observed cloud optical depth assuming 100% cloud coverage. For Casper site, the average reduction in local SW flux is 379W/m$^2$ or 50% and average reduction in global surface SW flux is 8%. For Columbia site, the average reduction in local surface SW flux is 329W/m$^2$ or 46% and average reduction in global surface SW flux is 7%.**

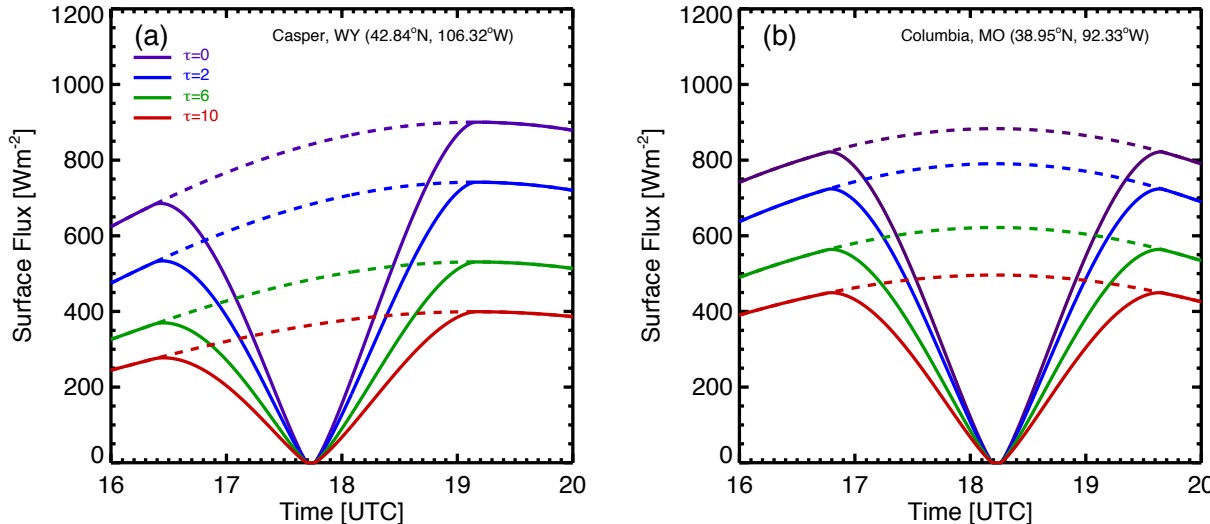

**Figure 8: (a) Casper, (b) Columbia. The modelled surface SW flux variations for eclipse (solid lines) and non-eclipse conditions for different cloud optical depth.**


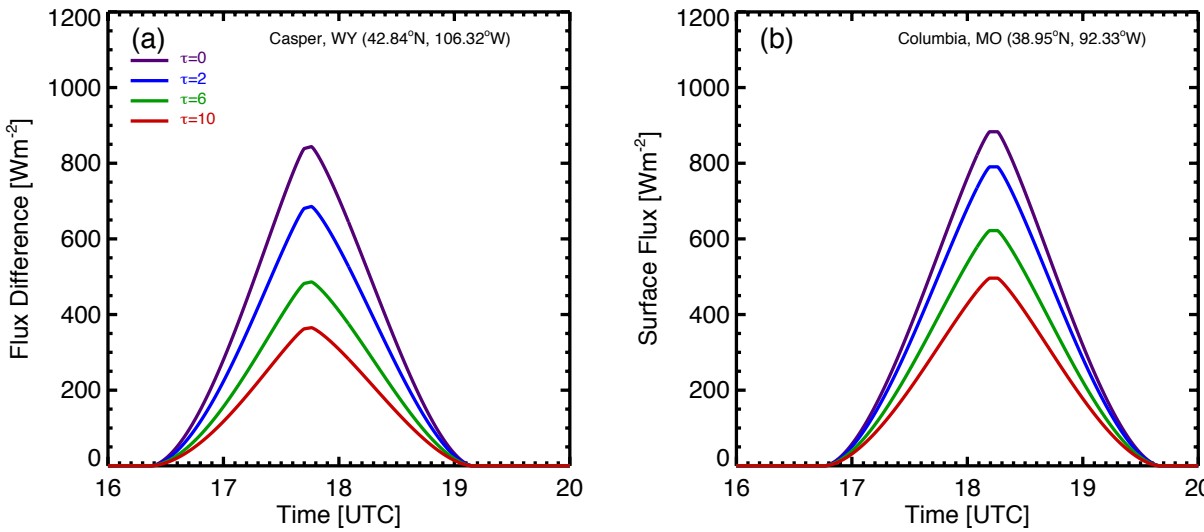


**Figure 9:** (a) Casper, (b) Columbia. The modelled surface SW flux reduction ($F_{non-eclipse,model} - F_{eclipse,model}$) for eclipse (solid lines) and non-eclipse conditions for different cloud optical depth.

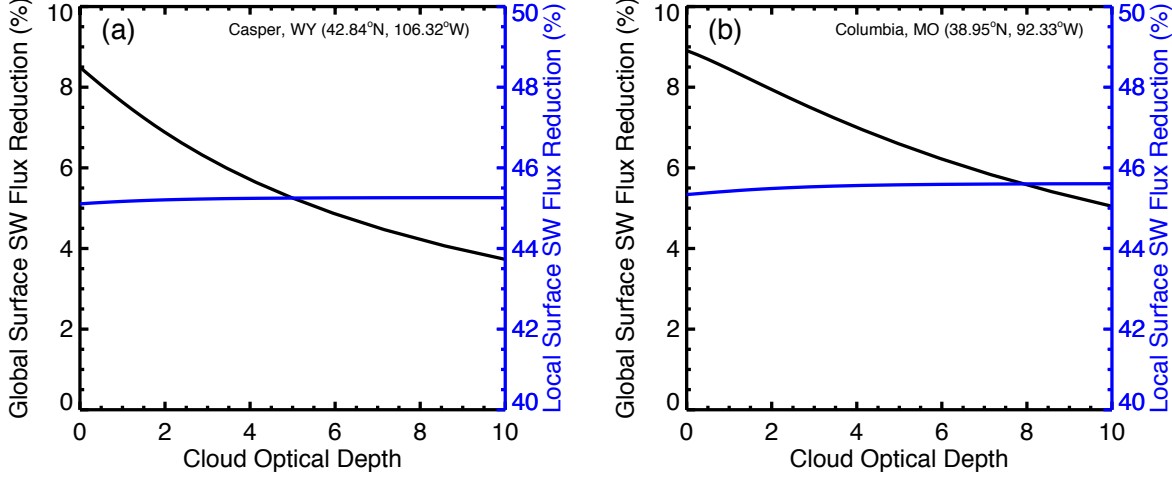


**Figure 10:** (a) Casper, (b) Columbia. The modelled relative reduction of average local surface flux (blue) during the eclipse and estimated impact on global surface SW flux budget (black).





**Table 1. Parameters for 21 August 2017 eclipse for Casper, Wyoming and Columbia Missouri. The first contact (C1), the moment when the Moon first touches the Sun's disk or the beginning of the partial eclipse; the second contact (C2), the beginning of totality; the maximum of the totality (Max); the third contact (C3), the end of totality; the fourth contact (C4), the instant when the Moon just leaves the Sun's disc or the end of the partial eclipse. The elevation of the site (Elev.) and solar zenith angle (SZA) and solar azimuth angle (SAA) at the totality are indicated.**

| Casper, WY (42°50'24.0" N, 106°19'12.0" W) Elev. = 1560 m, SZA = 36°, SAA = 143° | | Columbia, MO (38°56'53" N, 92°19'36'" W) Elev. = 227 m, SZA = 27°, SAA = 181° | |
|---|---|---|---|
| Event | Time (UTC) | Event | Time (UTC) |
| C1 | 16:22:16.0 | C1 | 16:45:40.8 |
| C2 | 17:42:36.3 | C2 | 18:12:20.3 |
| Max | 17:43:49.3 | Max | 18:13:38.8 |
| C3 | 17:45:04.5 | C3 | 18:14:59.2 |
| C4 | 19:09:23.7 | C4 | 19:40:12.8 |


**Table 2. Atmospheric properties including aerosol optical depth (AOD), ozone column amount ($O_3$), precipitable water vapor amount ($H_2O$), cloud optical depth (COD), and cloud top pressure (CTP) for Casper and Columbia sites. Note precipitable water vapor amounts are from nearest AERONET stations at St. Louis University, MO (38°38.16′ N, 90°13.9°′ W) and Spoon Butte, WY (42°35.76′ N, 104°26.58′ W) for Columbia and Casper, respectively.**

| | Casper, WY | Columbia, MO | Instrument |
|---|---|---|---|
| AOD | 0.23 | 0.19 | PSI-ER |
| $O_3$ | 313 DU* | 283 DU** | *EPIC, **PSI |
| $H_2O$ | 1.4 cm | 4.2 cm | Cimel |
| COD | variable | variable | PSI-ER |
| CTP | 327 mb* | 225 mb** | *MODIS, **VIIRS |


