# Peer review of "Changes in Surface Broadband Shortwave Radiation Budget during the 2017 Eclipse"

_Atmospheric Chemistry and Physics, 2019_

## Referee Comment (RC1) · Anonymous Referee #1 · 3 Mar 2020

The manuscript by Wen et al. discusses observations with pyranometers during the total solar eclipse of 21 August 2017. Pyranometer were located at two locations in the path of totality, at Casper, Wyoming, and Columbia, Missouri. Both locations were affected by clouds during the period of the eclipse. From their data, the authors reconstruct clear-sky measurements, i.e., they predict a time series of short wave (SW) irradiance at the two sites if there had been no clouds. In addition, they calculate the average reduction in SW irradiance for the two sites and the reduction of SW irradiance received by the Earth as a whole. The paper provides a quantification of the change in SW irradiance at two locations along the path of the solar eclipse of 21 August 2017. However, the findings are not generalized to be useful for the interpretation of the effect of solar eclipses in general. The topic of the paper is appropriate for publication in ACP.

[Figure]

General remarks

My main criticism of the paper is that the authors make many assumptions to simplify the problem at hand without estimating the impact of those assumptions on their results. Their findings are also not complemented with an uncertainty budget. For example, in Section 4.2., the effect of the eclipse on the reduction of global average irradiance is calculated based on the average SW flux within the area of the penumbral shadow "projected on Earth cross-section perpendicular to Sun-Earth line." According to Figure 6, a part of the perimeter of this area is outside the Earth. This circumstance is not even mentioned in the manuscript and will lead to errors when calculating the eclipse-induced relative reduction in SW flux according to Eq. (8.2) as changes in radiation outside the Earth's cross section are obviously inconsequential for the amount of energy received by Earth's surface.

Furthermore, the authors estimate the global surface effect of the eclipse from measurements at only two stations and do not take into account that the irradiance at Earth's surface does not only depend on the top of the atmosphere (TOA) reduction of irradiance resulting from the Moon's shadow but also on the path length of radiation traveling through the atmosphere. By basing their estimate only on two sites located in the path of totality, they neglect the fact that less radiation penetrates the atmosphere (both with and without eclipse) at high latitudes due to larger solar zenith angles (SZAs). Specifically, the authors assume in Eq. 8.3b that DeltaF is independent of azimuth angle phi but do not provide an estimate of the uncertainty caused by this assumption. While the effects of the difference in optical path at high and low latitudes partially cancel, the cancellation is not perfect because the change in atmospheric transmission depends exponentially on the optical path. The resulting uncertainty should be quantified.

In addition, the authors assert that the "temporal average of the observed surface SW flux from a local site is approximately equal to the spatial average of the surface SW flux" but do not try to estimate the uncertainty of this assumption. For example, uncertainties arise because the eclipse is not symmetric with respect to time. This is evident from Table 1, which shows that at Casper, the time difference between the first contact and totality is 1:21:33 while the difference between totality and forth contact is 1:25:34 - a difference of about 4 minutes. The effect may be small, but it should be quantified.

Lastly, estimating cloud optical depth from an instrument observing the direct Sun, like PANDORA, can lead to large errors, which are not discussed by the authors.

The PANDORA instrument has a relatively large field of view (FOV) of 2.2° (L101). This should work well for aerosol optical depth (AOD) retrievals if AODs are small, but could become problematic for estimating cloud optical depth, which are much larger. Ideally, the FOV of a radiometer looking at the direct beam for measuring optical depth should only be as large as the angular diameter of the Sun ($\sim$0.5°). (Of course, such a small FOV is not feasible due to tracking errors.) For large optical depths (e.g., tau > 4), the disk of the Sun is no longer clearly visible, and the radiation across the instrument's FOV is nearly uniform. In this case, the fraction of the instrument's signal that is contributed from the solid angle that contains the solar disk is only about (0.5°)^2 / (2.2°)^2 = 0.052 or 5.2%. As a consequence, the instrument "sees" much more light that it should and the resulting optical depth will be too small.

I suspect that the difference between the measured (green) and modeled (red) lines in Fig. 7 could be explained by values of cloud optical depth used in the model that are too small. If larger cloud optical depth were used, measurement and model should agree much better. The authors should provide an uncertainty estimate of the PANDORA-derived cloud optical depths and if necessary, apply a correction for the FOV effect. If my suspicion that systematic errors in the PANDORA cloud optical depth retrievals cause most of the difference between the measured and modeled results, the alternative explanation on line 356 ("Again, the cloud inhomogeneity is the main cause of the overestimation") may not be the dominating factor.

Specific comments:

L23: The sentence "The eclipse has a smaller impact on absolute value of surface flux reduction for cloudy conditions than a clear atmosphere; the impact decreases with the increase of cloud optical depth." is trivial and could be deleted (see also my comment with respect to line 364 below).

L78: The Sun was not "nearly overhead" at the two sites. According to Table 1, the SZAs at the time of totality were $36°$ and $27°$.

L119: Delete "collimated"

L144: The link http://aa.usno.navy.mil/data/docs/Eclipse2017.php does not work

L146: "the fact that the value of distance linearly correlated to time" is not a fact but a crude assumption. If that were the case, the time between 1st contact and totality and totality and 4th contact would be the same, but it is not.

Eq. (1) and Figure 2: Beer-Lambert's law only applies to monochromatic radiation. What wavelength is discussed here?

L169: irradiance > spectral irradiance at a wavelength of xxx nm

L200 - 201: This paragraph is a misrepresentation of the paper by Koepke et al. (2001) and focuses on a small detail of that paper. I would say: Amongst others, Koepke et al. (2001) estimated ..." Also, change "normalized radiance" to "the ratio of spectral irradiance at 310 nm calculated for eclipse and non-eclipse conditions".

L202: delete "radiance and". (Radiance (e.g., from the sky) are neither discussed by Koepke or in this paper).

L266 - 270: The value of DeltaF depends greatly on the cloud condition at the time of the eclipse. If I understand the text correctly, F2 (i.e., the denominator of Eq. (8.2)) was multiplied by factor of 0.55 to account for the average global transmission of the atmosphere. If clouds and other atmospheric absorbers and scatterers within the area of the eclipse would attenuate the TOA irradiance by the same factor, Delta F would

provide a good estimate of the "mean" global consequences of the eclipse. However, if the area affected by the eclipse were either clear-sky or shrouded by an optical thick cloud, DeltaF could greatly deviate (either up or down) from this average value. While this issue is discussed to some extend later when results for Casper and Columbia are presented, the limitations of estimating the global consequences of an eclipse should already be introduced here. The authors should keep in mind that their paper would be more useful if their results could also be applied to future eclipses occurring at different locations and cloud conditions. A generalization of their findings would be appreciated by readers.

L271: The link http://aa.usno.navy.mil/data/docs/geocentric.php does not work.

L280: In line 259, F_eclipse was defined as the average flux in the 2D area of the Moon's shadow. Here F_eclipse is defined as the average of F along the totality path. This is not the same. The equation should be deleted here because only the definition on L259 (and the calculations in Eq. (8.3)) are relevant here.

L292: As mentioned in my general comment, the assumption that fluxes at Earth's surface are independent of the azimuth angle is rather crude and the uncertainty of this assumption should be quantified.

L296-297: Change "F_non-eclipse(r) is derived from F_non-eclipse(r) (Eq. 6.2)." to "F_non-eclipse(r) is derived from F_eclipse(r) according to Eq. (6.2)." (Note that the original sentence includes "non" twice.)

L303: Why does more cloud cover lead to more realistic atmospheric conditions? Do you mean that the conditions at Columbia are more representative for the average attenuation of radiation, e.g., as expressed earlier by the factor of 0.55?

L314-317 & L324: While it is true that clouds can lead to irradiances at the ground exceeding the clear-sky limit, an enhancement of about 22% near the time of totality (as indicated in Fig. 7a) that is caused by "thin cirrus clouds" seems to be rather

large. I suspect that systematic errors in the conversion of the observed measurements (black line in Fig. 7a) to the "derived non-eclipsed" dataset (green line) may have also contributed to this large enhancement. The uncertainty of this conversion should be given or at least it should be acknowledged that systematic errors in the conversion could have contributed to the large enhancement effect.

L330: As mentioned earlier, I suspect that systematic errors in the cloud optical depth data from the PANDORA spectrometer are the main cause of the discrepancies between the modeled (red) and derived (green) surface SW irradiance at time where clouds attenuate (i.e., when the Sun is behind the cloud). The authors should quantify systematic errors in PANDORA-derived cloud optical depths.

L356: The sentence "the cloud inhomogeneity is the main cause of the overestimation" is just an assertion without basis. As mentioned above, I think the large FOV of the PANDORA spectroradiometer is the main cause of the overestimation. If the authors disagree, they should provide quantitative evidence that cloud inhomogeneity is really the main culprit.

L364-375, Figure 8, Figure 9: It is rather trivial that the effect of the eclipse leads to smaller *absolute* changes during cloudy than clear conditions. I am not sure why this is discussed in such detail here, and even mentioned in the abstract.

L380: Please delete "rigorously". As mentioned above, there are many assumptions and simplifications going in these calculations. I would not consider them "rigorous".

L382-393, Figure 10: My take-home message from this paragraph and the figure is that the global effect of an eclipse on SW irradiance is between about 4 and 10%, and depends on a lot of factors, including SZA and cloud optical depth. Such a wide range of reductions is not very useful. It would be nice if the authors could generalize their results to make them more applicable for other eclipses.

L409: I disagree with the conclusions that "clouds play a unique role in modifying the

[Figure]

surface flux reduction during an eclipse". As correctly concluded in the paper, clouds attenuate the incoming radiation by about the same percentage during a partial eclipse and during a normal day, except of the red-shift effect (Fig. 5), which is smaller than 5%. So I don't see anything "unique" about clouds (with the exception that they are a nuisance when interpreting measurements during an eclipse.)

Figure 2: Specify wavelength. In the caption, change "radiances" to "irradiance" (The optical depth refers to the attenuation of the direct solar beam.)

Figure 3: Explain color scale of panels (d) and (e).

Figure 4: Change "nearest station in Springfied" to "Springfield, the nearest station to Columbia".

Figure 5: The font size is the insert of panel (a) is too small. Define the term "spectral transmittance" (make clear that transmittance refers to the global (sun and sky) irradiance at the surface, not just the solar beam).

Technical corrections:

L23: on absolute > on the absolute

L34: arctic > Arctic

L192: error cloud inhomogeneity > error in cloud inhomogeneity

L230: slight > slightly

L368: optical depth > optical depths

L473: The paper by Ockenfuß is now published.

———————————————————

---

## Author Comment (AC1) · 18 Mar 2020

The manuscript by Wen et al. discusses observations with pyranometers during the total solar eclipse of 21 August 2017. Pyranometer were located at two locations in the path of totality, at Casper, Wyoming, and Columbia, Missouri. Both locations were affected by clouds during the period of the eclipse. From their data, the authors re-construct clear-sky measurements, i.e., they predict a time series of short wave (SW) irradiance at the two sites if there had been no clouds. In addition, they calculate the

average reduction in SW irradiance for the two sites and the reduction of SW irradiance received by the Earth as a whole. The paper provides a quantification of the change in SW irradiance at two locations along the path of the solar eclipse of 21 August 2017. However, the findings are not generalized to be useful for the interpretation of the effect of solar eclipses in general. The topic of the paper is appropriate for publication in ACP.

Reply: We thank the reviewer for taking time to review our paper and making constructive comments, particularly some details we were not aware of, that really helped us to improve our manuscript. However, it seems that the reviewer missed the major contribution of this paper, that is that we found that "the non-eclipse-to-eclipse surface flux ratio depends strongly on the obscuration of solar disk and slightly on cloud optical depth. These findings allowed us to estimate what the surface broadband SW flux would be for non-eclipse conditions from observations during the eclipse and further to quantify the impact of the eclipse on the surface broadband SW radiation budget." and "the relative time-averaged reduction of local surface SW flux during a solar eclipse is approximately 45% and it is not sensitive to cloud optical depth." (see abstract and text) These findings are general and can be used for estimating SW flux for hypothetical non-eclipse conditions during solar eclipse in the past and future.

General remarks My main criticism of the paper is that the authors make many assumptions to simplify the problem at hand without estimating the impact of those assumptions on their results. Their findings are also not complemented with an uncertainty budget. For example, in Section 4.2., the effect of the eclipse on the reduction of global average irradiance is calculated based on the average SW flux within the area of the penumbral shadow "projected on Earth cross-section perpendicular to Sun-Earth line." According to Figure 6, a part of the perimeter of this area is outside the Earth. This circumstance is not even mentioned in the manuscript and will lead to errors when calculating the eclipse-induced relative reduction in SW flux according to Eq. (8.2) as changes in radiation outside the Earth's cross section are obviously inconsequential for the amount of energy received by Earth's surface.

Reply: We agree with the reviewer about the importance of estimating uncertainties. We have specified instrument errors in section 2. From theoretical calculation we estimated the error in derived non-eclipse surface shortwave flux is less 4% (see lines 253-256). About "a part of the perimeter of this area is outside the Earth", we recalculated the irradiance just in the Moon's shadow area projected on the Earth's disk (see Eqs 9.1, 9.2, 10.2). The area of the Moon's shadow on the Earth's disk is $0.91*Pi*r0^2$ and $0.97*Pi*r0^2$ when Casper and Columbia sites were experiencing totality (see lines 288 on page 10). As a result, the global average surface flux reduction is changed from 8% in previous version to 7.4% in the current version of the paper when the totality was Casper and from 7% to 6.8% (see lines 351 and 376).

Furthermore, the authors estimate the global surface effect of the eclipse from measurements at only two stations and do not take into account that the irradiance at Earth's surface does not only depend on the top of the atmosphere (TOA) reduction of irradiance resulting from the Moon's shadow but also on the path length of radiation traveling through the atmosphere. By basing their estimate only on two sites located in the path of totality, they neglect the fact that less radiation penetrates the atmosphere (both with and without eclipse) at high latitudes due to larger solar zenith angles (SZAs). Specifically, the authors assume in Eq. 8.3b that DeltaF is independent of azimuth angle phi but do not provide an estimate of the uncertainty caused by this assumption. While the effects of the difference in optical path at high and low latitudes partially cancel, the cancellation is not perfect because the change in atmospheric transmission depends exponentially on the optical path. The resulting uncertainty should be quantified.

Reply: We agree with reviewer's comment about estimating the uncertainty of the assumption. In this case, we couldn't provide a reasonable estimate since lack of full 3D cloud property information. About modeling the true radiative transfer for eclipse conditions, Emde and Mayer (2007) made wonderful statement: "the full characterization of the input parameters (cloud properties) with high enough accuracy to actually

constrain the model result is close to impossible." Thus, to quantify the uncertainty is also close to impossible. However, we consider the results as a first order estimate of global average surface SW flux reduction for the conditions at the two sites. And it is useful for understand the radiative processes during eclipse. As far as we know, this is the first work to quantify the impact of eclipse on surface SW flux and hope it could motivate more research activity in this small area.

In addition, the authors assert that the "temporal average of the observed surface SW flux from a local site is approximately equal to the spatial average of the surface SW flux" but do not try to estimate the uncertainty of this assumption. For example, uncertainties arise because the eclipse is not symmetric with respect to time. This is evident from Table 1, which shows that at Casper, the time difference between the first contact and totality is 1:21:33 while the difference between totality and forth contact is 1:25:34 - a difference of about 4 minutes. The effect may be small, but it should be quantified.

Reply: According to Koepke et al. (2001) (cited in line 146, on page 5 in our paper), "The value of X is linearly correlated to time." (in Section 2 of their paper) (X is Sun-Moon distance) and linearly translate between X and distance in the reference plane is a common practice in computing radiative properties under eclipse conditions (see Emde and Mayer 2007). Thus, the linear relation between the distance and time is valid. In the revised version, we emphasize that "we estimate r from the linear relation with t for the time periods before and after the totality separately because of the asymmetry of the two branches." (lines 313-314 on page 11)

Lastly, estimating cloud optical depth from an instrument observing the direct Sun, like PANDORA, can lead to large errors, which are not discussed by the authors. The PANDORA instrument has a relatively large field of view (FOV) of 2.2_ (L101). This should work well for aerosol optical depth (AOD) retrievals if AODs are small, but could become problematic for estimating cloud optical depth, which are much larger. Ideally, the FOV of a radiometer looking at the direct beam for measuring optical depth should only be as large as the angular diameter of the Sun (_0.5_). (Of course, such

a small FOV is not feasible due to tracking errors.) For large optical depths (e.g., tau > 4), the disk of the Sun is no longer clearly visible, and the radiation across the instrument's FOV is nearly uniform. In this case, the fraction of the instrument's signal that is contributed from the solid angle that contains the solar disk is only about (0.5_)ËĘ2 / (2.2_)ËĘ2 = 0.052 or 5.2%. As a consequence, the instrument "sees" much more light that it should and the resulting optical depth will be too small. I suspect that the difference between the measured (green) and modeled (red) lines in Fig. 7 could be explained by values of cloud optical depth used in the model that are too small. If larger cloud optical depth were used, measurement and model should agree much better. The authors should provide an uncertainty estimate of the PANDORA derived cloud optical depths and if necessary, apply a correction for the FOV effect. If my suspicion that systematic errors in the PANDORA cloud optical depth retrievals cause most of the difference between the measured and modeled results, the alternative explanation on line 356 ("Again, the cloud inhomogeneity is the main cause of the overestimation") may not be the dominating factor.

Reply: This is a very good question. We performed a series radiative transfer calculation. We found (1) the error for AOD is negligible (this was also reported by Sinyuk et al. 2012); (2) the error for water cloud is larger than error for aerosol, but it less than 5% for cloud optical depth less than 6; (3) the error for ice cloud is large due to strong forward peak of ice crystal. In this revised version we corrected cirrus cloud optical depth (new Figure 2) and the corrected cloud optical depths were used in radiative transfer calculations (new Figure 7). The modeled shortwave flux decreased compared to the results in the previous version as the reviewer suggested. For the Columbia site, the 1D modeled surface SW flux matched the derived one at ∼17.65 UTC and ∼18.65 UTC. However, the major feature remains the same. Our conclusion stands. This is not a surprise because of the inhomogeneity nature of cloud. The cloud inhomogeneity also can be seen from the variation of the corrected atmospheric optical depths in Figure 2. We do appreciate your comments that motivated us to look into this problem in more details.

Specific comments: L23: The sentence "The eclipse has a smaller impact on absolute value of surface flux reduction for cloudy conditions than a clear atmosphere; the impact decreases with the increase of cloud optical depth." is trivial and could be deleted (see also my comment with respect to line 364 below). Reply: You are right hat it is trivial. We decided to mention it because we wanted to address the results in the sentence followed "the relative reduction is not sensitive to cloud optical depth", which was not so trivial in the beginning.

L78: The Sun was not "nearly overhead" at the two sites. According to Table 1, the SZAs at the time of totality were 36_ and 27_. Reply: Changed "nearly overhead" to "nearly noon"

L119: Delete "collimated" Done

L144: The link http://aa.usno.navy.mil/data/docs/Eclipse2017.php does not work Reply: Their website (https://www.usno.navy.mil/USNO) says "The USNO websites http://aa.usno.navy.mil/ (and a few other sites) are undergoing modernization efforts. The expected completion of the work and the estimated return of service is Summer 2020). I have to use the data Fred Espenak provided to me. And the data files are in supplement material.

L146: "the fact that the value of distance linearly correlated to time" is not a fact but a crude assumption. If that were the case, the time between 1st contact and totality and totality and 4th contact would be the same, but it is not. Eq. (1) and Figure 2: Beer-Lambert's law only applies to monochromatic radiation. What wavelength is discussed here? Reply: The linear relation between distance and time is not our invention. Koepke et al. (2001) stated that "The value of X is linearly correlated to time", where X is the magnitude of the eclipse. We confirmed that it is true for both Casper and Columbia sites, and the correlation coefficient of X and t is 0.999956 and 0.999996 for Casper and Columbia site, respective. We followed Emde and Mayer's (2007) procedure (linear translation between X and distance in the reference plane, see section

2.3 of their paper) to relate X to distance or time t to distance. About Eq. (1) and Figure 2, we added "spectral irradiance at 550 nm" (see line 159) and Figure 2 caption.

L169: irradiance > spectral irradiance at a wavelength of xxx nm Done

L200 - 201: This paragraph is a misrepresentation of the paper by Koepke et al. (2001) and focuses on a small detail of that paper. I would say: Amongst others, Koepke et al. (2001) estimated ..." Also, change "normalized radiance" to "the ratio of spectral irradiance at 310 nm calculated for eclipse and non-eclipse conditions". Reply: The part Koepke's work we referenced (using the ratio to estimate the photolysis frequencies for non-eclipse conditions using observed photolysis frequencies) is critical for us to develop our method to estimate the surface SW flux under non-eclipse conditions. See section 4.1 for detail.

L202: delete "radiance and". (Radiance (e.g., from the sky) are neither discussed by Koepke or in this paper). Reply: Though Koepke didn't discuss radiance or irradiance, Eq. (3.1) in their paper holds for both spectral radiance and irradiance for a reduced TOA spectral irradiance during eclipse because the transmittance remains the same for eclipse and hypothetical non-eclipse conditions.

L266 - 270: The value of DeltaF depends greatly on the cloud condition at the time of the eclipse. If I understand the text correctly, F2 (i.e., the denominator of Eq. (8.2)) was multiplied by factor of 0.55 to account for the average global transmission of the atmosphere. If clouds and other atmospheric absorbers and scatterers within the area of the eclipse would attenuate the TOA irradiance by the same factor, Delta F would provide a good estimate of the "mean" global consequences of the eclipse. However, if the area affected by the eclipse were either clear-sky or shrouded by an optical thick cloud, DeltaF could greatly deviate (either up or down) from this average value. While this issue is discussed to some extend later when results for Casper and Columbia are presented, the limitations of estimating the global consequences of an eclipse should already be introduced here. The authors should keep in mind that their paper would be

more useful if their results could also be applied to future eclipses occurring at different locations and cloud conditions. A generalization of their findings would be appreciated by readers. Reply: We didn't claim the results are general for any locations on the Earth. We emphasize that the temporal average value from one location represents the spatial average for similar atmosphere and surface conditions in the penumbra. The results from the Casper site represent mostly clear atmospheric condition. With more cloud cover over the Columbia site, the estimated shortwave irradiance change is closer to realistic atmospheric condition as described later. But the method to estimate SW flux under hypothetical non-eclipse condition during a solar eclipse we developed in this work is general and can be applied to any solar eclipse event in the past and future. Basically, it is Eq. (8.2)

L271: The link http://aa.usno.navy.mil/data/docs/geocentric.php does not work. Reply: Their website (https://www.usno.navy.mil/USNO) says "The USNO websites http://aa.usno.navy.mil/ (and a few other sites) are undergoing modernization efforts. The expected completion of the work and the estimated return of service is Summer 2020). The parameters from the website are inserted into the text (see line 285). And that link is removed.

L280: In line 259, F_eclipse was defined as the average flux in the 2D area of the Moon's shadow. Here F_eclipse is defined as the average of F along the totality path. This is not the same. The equation should be deleted here because only the definition. on L259 (and the calculations in Eq. (8.3)) are relevant here. Reply: Good point. But we feel it is easier to follow from simple to more complex situation.

L292: As mentioned in my general comment, the assumption that fluxes at Earth's surface are independent of the azimuth angle is rather crude and the uncertainty of this assumption should be quantified. Reply: Again, we could not provide an estimate of the uncertainty caused by this assumption since we don't have 3D cloud property information and computationally almost impossible to simulate the truth in such large area. As mentioned earlier, Emde and Mayer (2007) made wonderful statement: "the

full characterization of the input parameters (cloud properties) with high enough accuracy to actually constrain the model result is close to impossible." Thus, to quantify the uncertainty is also close to impossible. We consider the results as a first order estimate of global average surface SW flux reduction for the conditions at the two sites. This is the first time, as we know so far, the impact of solar eclipse on surface SW flux is quantified. The results are useful for understand the radiative transfer processes during eclipse. We do hope our results could help researcher do a better job in the future.

L296-297: Change "F_non-eclipse(r) is derived from F_non-eclipse(r) (Eq. 6.2)." to "F_non-eclipse(r) is derived from F_eclipse(r) according to Eq. (6.2)." (Note that the original sentence includes "non" twice.) Fixed

L303: Why does more cloud cover lead to more realistic atmospheric conditions? Do you mean that the conditions at Columbia are more representative for the average attenuation of radiation, e.g., as expressed earlier by the factor of 0.55? Reply: We mean that "The results from the Casper site represent mostly clear atmospheric condition." Since the Earth's cloud cover is about 0.68 for cloud optical depth larger than 0.1, one experiences cloudy atmosphere more than clear atmosphere. With more cloud cover, the atmospheric condition in Columbia is closer to realistic atmospheric condition.

L314-317 & L324: While it is true that clouds can lead to irradiances at the ground exceeding the clear-sky limit, an enhancement of about 22% near the time of totality (as indicated in Fig. 7a) that is caused by "thin cirrus clouds" seems to be rather large. I suspect that systematic errors in the conversion of the observed measurements (black line in Fig. 7a) to the "derived non-eclipsed" dataset (green line) may have also contributed to this large enhancement. The uncertainty of this conversion should be given or at least it should be acknowledged that systematic errors in the conversion could have contributed to the large enhancement effect. Reply: 22% enhancement is not a surprise. We did a detailed analysis. We found that thin cirrus cloud (not in the in the line between the Sun and ground-based instrument) with optical depth of 0.25 and

cloud fraction of 0.2 can lead to an enhancement of ~30W/m2 just before the eclipse started due the diffuse irradiance generated by the cirrus. In optically thin regime, SW diffuse flux is approximately linearly related to cloud optical depth. That means that thin cirrus cloud (not in the line between the Sun and the instrument) with optical depth of 1 and cloud fraction of 0.4 can generate extra 240W/m2. Remember that the error in the conversion is about 4%.

L330: As mentioned earlier, I suspect that systematic errors in the cloud optical depth data from the PANDORA spectrometer are the main cause of the discrepancies between the modeled (red) and derived (green) surface SW irradiance at time where clouds attenuate (i.e., when the Sun is behind the cloud). The authors should quantify systematic errors in PANDORA-derived cloud optical depths. Reply: Cloud optical depths have been corrected. Surface SW flux have been recalculated. (see reply to the major comments).

L356: The sentence "the cloud inhomogeneity is the main cause of the overestimation" is just an assertion without basis. As mentioned above, I think the large FOV of the PANDORA spectroradiometer is the main cause of the overestimation. If the authors disagree, they should provide quantitative evidence that cloud inhomogeneity is really the main culprit. Reply: Again, cloud optical depths have been corrected. Surface SW flux have been recalculated. (see reply to the major comment).

L364-375, Figure 8, Figure 9: It is rather trivial that the effect of the eclipse leads to smaller *absolute* changes during cloudy than clear conditions. I am not sure why this is discussed in such detail here, and even mentioned in the abstract. Reply: It may seem trivial to one who has some experience on this subject. It may not rather trivial to others. Besides, this topic has never been discussed, as far as we know, in any journal. Thus, this is an important point of this paper.

L380: Please delete "rigorously". As mentioned above, there are many assumptions and simplifications going in these calculations. I would not consider them "rigorous".

Done

L382-393, Figure 10: My take-home message from this paragraph and the figure is that the global effect of an eclipse on SW irradiance is between about 4 and 10%, and depends on a lot of factors, including SZA and cloud optical depth. Such a wide range of reductions is not very useful. It would be nice if the authors could generalize their results to make them more applicable for other eclipses. Reply: Although we study surface SW flux only for 2017 solar eclipse, one conclusion is general for other eclipse. That is "the relative time-averaged reduction of local surface SW flux during a solar eclipse is approximately 45% and it is not sensitive to cloud optical depth." The method to derive surface SW flux under hypothetical non-eclipse conditions is also general. This is stated in the abstract.

L409: I disagree with the conclusions that "clouds play a unique role in modifying the surface flux reduction during an eclipse". As correctly concluded in the paper, clouds attenuate the incoming radiation by about the same percentage during a partial eclipse and during a normal day, except of the red-shift effect (Fig. 5), which is smaller than 5%. So I don't see anything "unique" about clouds (with the exception that they are a nuisance when interpreting measurements during an eclipse.) Relpy: We performed a details spectral radiative transfer calculation and made that conclusion. In lines 235-240, "Fig. 5(b) shows that an increase of cloud optical depth leads to a relatively larger decrease of surface spectral irradiance in near-IR wavelengths compared to near-UV and visible wavelengths. Here we examine the effect of cloud on transmitted flux for red-shift spectral solar irradiance. For the red-shift spectrum, an increase in cloud optical depth leads to a relatively smaller decrease in transmitted surface flux in near-UV and visible wavelengths. There is a relatively larger decrease in near-IR wavelengths compared to the spectrum for the normal conditions simply because of the red-shift in TOA solar spectrum. To some extent, the larger decrease in near-IR wavelengths compensates for the smaller decrease in visible and near-UV wavelengths, resulting in a decrease in spectrally integrated surface SW flux similar to that for the normal TOA

[Figure]

spectral solar irradiance." (Basically the unique feature of cloud absorption leads to a compensation for spectrally integrated surface irradiance for the red-shifted spectral solar irradiance during eclipse, resulting a small dependence on cloud optical depth.) This allowed us to derived surface SW flux under hypothetical non-eclipse conditions.

Figure 2: Specify wavelength. In the caption, change "radiances" to "irradiance" (The optical depth refers to the attenuation of the direct solar beam.) Done Figure 3: Explain color scale of panels (d) and (e). Done. See Figure 3 caption. Figure 4: Change "nearest station in Springfied" to "Springfield, the nearest station to Columbia". Done Figure 5: The font size is the insert of panel (a) is too small. Define the term "spectral transmittance" (make clear that transmittance refers to the global (sun and sky) irradiance at the surface, not just the solar beam). Reply: defined now (see line 234). Technical corrections: L23: on absolute > on the absolute Done L34: arctic > Arctic Done L192: error cloud inhomogeneity > error in cloud inhomogeneity Done L230: slight > slightly Done L368: optical depth > optical depths Done L473: The paper by Ockenfuß is now published. Done

Please also note the supplement to this comment:
https://www.atmos-chem-phys-discuss.net/acp-2019-961/acp-2019-961-AC1-supplement.zip
* * *
[Figure]

[Figure]

**Fig. 1.**

[Figure]

[Figure]

**Fig. 2.**

**(a)** Casper, WY (42.84°N, 106.32°W)
Observation
Derived Non-eclipse
Model Non-eclipse cloudy

Surface Flux [Wm⁻²] vs Time [UTC]

Clear Eclipse
Clear Non-eclipse

**Fig. 3.**

[Figure]

**Fig. 4.**

---

## Referee Comment (RC2) · Anonymous Referee #3 · 2 Jul 2020

The authors conducted a ground-based experiment to observe broadband shortwave irradiance at Casper, Wyoming and Columbia, Missouri, located in the totality path of the August 2017 solar eclipse but separated by 1200 km. Surface shortwave flux measurements with simultaneous atmospheric observations allow the investigation of the impact of the solar eclipse on the surface shortwave radiative budget under different atmospheric conditions. Radiative transfer calculations show that the non-eclipse-to-eclipse surface SW flux ratio primarily depends on the obscuration of the solar disk during eclipse and slightly depends on cloud optical depth. The noneclipse surface SW flux can then be derived by multiplying the observed SW flux with the modelled surface SW flux ratio. It was found that at the Casper site, the eclipse led to a decrease of 379 W m-2 (50%) in averaged local surface SW flux, and the Moon's shadow caused

about a 8% reduction in global average surface SW radiation budget when the totality was at Casper; at the Columbia site, the eclipse led to a decrease of 329 W m-2 (46%) in averaged local surface SW flux, and the Moon's shadow caused about a 7% reduction in global average surface SW radiation budget when the totality was at Columbia. The paper is well written and adds useful information to the existing literature on the impact of the eclipse on the surface broadband shortwave radiation budget. I recommend the paper be accepted by the ACP after addressing the following comments and suggestions (which follow the order of the presentation instead of the importance).

1. 80: Some information on the spectral resolution of the PSI and PSI-ER would be useful.

2. 90: The TDE correction is 18 W/m2 at Casper but only 2 W/m2 at Columbia at the totality. Any explanation about such difference?

3. 105: What is the wavelength for the aerosol optical depth listed in Table 2?

4. Clarify that in Eq. (1) the "I" is the solar monochromatic direct irradiance.

5. 179: What are the wavelengths here? Is the gaseous absorption negligible?

6. 205: Consider to add the subscript "o" for "I" in Eqs. (4) and (5) to be separated from the "I" at the surface in Eq. (1).

7. Equation (6.1) makes an important assumption that the non-eclipse-to-eclipse surface SW flux ratio for realistic 3D cloudy atmospheric conditions is approximately equal to the 1D model computed flux ratio for clear atmospheric conditions. Some discussions on the potential impact of this assumption are necessary.

8. Consider to use "RDF" or "RD" instead of "deltaF" in Eqs. (8.1), (8.2), and 8.3) for the relative differences.

---

## Author Response (AR1)

The authors conducted a ground-based experiment to observe broadband shortwave irradiance at Casper, Wyoming and Columbia, Missouri, located in the totality path of the August 2017 solar eclipse but separated by 1200 km. Surface shortwave flux measurements with simultaneous atmospheric observations allow the investigation of the impact of the solar eclipse on the surface shortwave radiative budget under different atmospheric conditions. Radiative transfer calculations show that the non-eclipse-to eclipse surface SW flux ratio primarily depends on the obscuration of the solar disk during eclipse and slightly depends on cloud optical depth. The noneclipse surface SW flux can then be derived by multiplying the observed SW flux with the modelled surface SW flux ratio. It was found that at the Casper site, the eclipse led to a decrease of 379 W m-2 (50%) in averaged local surface SW flux, and the Moon's shadow caused about a 8% reduction in global average surface SW radiation budget when the totality was at Casper; at the Columbia site, the eclipse led to a decrease of 329 W m-2 (46%) in averaged local surface SW flux, and the Moon's shadow caused about a 7% reduction in global average surface SW radiation budget when the totality was at Columbia. The paper is well written and adds useful information to the existing literature on the impact of the eclipse on the surface broadband shortwave radiation budget. I recommend the paper be accepted by the ACP after addressing the following comments and suggestions (which follow the order of the presentation instead of the importance).

*We thank the reviewer for taking time to review our paper during the pandemic crisis and making constructive comments that really helped us to improve our manuscript. We have revised our manuscript based on the reviewer's comments and suggestions.*

1. 80: Some information on the spectral resolution of the PSI and PSI-ER would be useful.

*Now they are specified (see lines 83-85).*

2. 90: The TDE correction is 18 W/m2 at Casper but only 2 W/m2 at Columbia at the totality. Any explanation about such difference?

*Since this paper already covered a broad topic related to SW radiation budget, the method of TDE correction was cited directly to peer-reviewed papers (e.g., Ji et al., 2011) without explicitly shown the equations. From Ji et al. (2011), the TDE term is calculated from differences between the temperature of thermopile and dome (f\*sigma\*(Ts^4 – Td^4), see Eq.2 of Ji et al.), where Ts and Td denote respectively the temperatures of thermopile and dome. The glass dome of PSP transits most of the SW radiation, but absorbs LW radiation. From our observations, there was more cloud coverage at Columbia site than that at Casper site; in turn, time series of Td were warmer at Columbia than at Casper due to the absorption of LW radiation emitted by clouds. Thus, the attached figure shows larger temperature differences of (Ts – Td) at Casper (for example, during the totality of ~2.2 degrees C at Casper vs. ~0.2 degrees C at Columbia) and resulted in larger TDE-correction term of f\*sigma\*(Ts^4 – Td^4).*

[Figure]

3. 105: What is the wavelength for the aerosol optical depth listed in Table 2?
*Aerosol optical depth is observed at 550 nm. This is clarified (see line 160 and Table 2).*

4. Clarify that in Eq. (1) the "I" is the solar monochromatic direct irradiance.
*This is clarified (see line157)*

5. 179: What are the wavelengths here? Is the gaseous absorption negligible?
*For PSI, the cloud optical depth is derived at 550 nm where the absorption of Chappuis ozone bands is negligible. Now the wavelength is specified (see line 191). The cloud top pressure is from MODIS L2 Collection 6 data. For mid and high cloud, the cloud top pressure is estimated using 15 μm band radiance, low level cloud top pressure is estimated from 11 μm window channel. A paper describing MODIS cloud top properties (Baum et al., 2012) is referenced.*

6. 205: Consider to add the subscript "o" for "I" in Eqs. (4) and (5) to be separated from the "I" at the surface in Eq. (1).
*Good suggestion. They have been modified in Eqs. (6) and (7) now.*

7. Equation (6.1) makes an important assumption that the non-eclipse-to-eclipse surface SW flux ratio for realistic 3D cloudy atmospheric conditions is approximately equal to the 1D model computed flux ratio for clear atmospheric conditions. Some discussions on the potential impact of this assumption are necessary.
*The 1D model introduces large error in umbra and bordering area. This is discussed in the revised version of the manuscript (see lines 269-279).*

8. Consider to use "RDF" or "RD" instead of "deltaF" in Eqs. (8.1), (8.2), and 8.3) for the relative differences.
*Now we use $\Delta F_r$ for relative differences now Eqs. (10.1), (10.2), and (10.3).*

[revised manuscript text omitted]